# Machine learning reveals distinct neuroanatomical signatures of cardiovascular and metabolic diseases in cognitively unimpaired individuals

Sindhuja Tirumalai Govindarajan [1] ✉, Elizabeth Mamourian [1], Guray Erus [1], Ahmed Abdulkadir[2], Randa Melhem[1], Jimit Doshi[1], Raymond Pomponio[1], Duygu Tosun [3], Murat Bilgel [4], Yang An[4], Aristeidis Sotiras [5], Daniel S. Marcus[5], Pamela LaMontagne [5], Tammie L. S. Benzinger[5], Mark A. Espeland[6,7], Colin L. Masters[8], Paul Maruff[8], Lenore J. Launer [9], Jurgen Fripp[10], Sterling C. Johnson [11], John C. Morris[12], Marilyn S. Albert[13], R. Nick Bryan[14], Susan M. Resnick [4], Mohamad Habes [15], Haochang Shou [1,16], David A. Wolk[17], Ilya M. Nasrallah [1,14] & Christos Davatzikos [1] ✉

Comorbid cardiovascular and metabolic risk factors (CVM) differentially impact brain structure and increase dementia risk, but their specific magnetic resonance imaging signatures (MRI) remain poorly characterized. To address this, we developed and validated machine learning models to quantify the distinct spatial patterns of atrophy and white matter hyperintensities related to hypertension, hyperlipidemia, smoking, obesity, and type-2 diabetes mellitus at the patient level. Using harmonized MRI data from 37,096 participants (45–85 years) in a large multinational dataset of 10 cohort studies, we generated five in silico severity markers that: i) outperformed conventional structural MRI markers with a ten-fold increase in effect sizes, ii) captured subtle patterns at sub-clinical CVM stages, iii) were most sensitive in mid-life (45–64 years), iv) were associated with brain beta-amyloid status, and v) showed stronger associations with cognitive performance than diagnostic CVM status. Integrating personalized measurements of CVM-specific brain signatures into phenotypic frameworks could guide early risk detection and stratification in clinical studies.

Age-related progressive accumulation of chronic and often co-occurring modifiable health risks is estimated to contribute up to 50% of all incident dementia cases globally[1], with population-attributable risks of 23.8% for hypertension[1], 14.1% for smoking[2], 20.9% for obesity, and 12.5% for type 2 diabetes[3]. The prevalence of these conditions and their contribution to dementia vary significantly by race, ethnicity, and socioeconomic status, resulting in health inequalities[3]. Robust detection of early neuroanatomical changes associated with these cardiovascular and metabolic risk factors (CVMs) could pave the pathway for early risk stratification, longitudinal monitoring of disease progression, and proactive management to mitigate cognitive decline. Understanding the distinct associations between specific CVMs and in vivo brain changes is crucial to disentangle the combined effects of comorbid CVMs and prioritize intervention targets.

A full list of affiliations appears at the end of the paper. ✉e-mail: sindhuja.tirumalaigovindarajan@pennmedicine.upenn.edu; Christos.davatzikos@pennmedicine.upenn.edu

Structural magnetic resonance imaging (sMRI) investigations based on diagnostic CVM labels have uncovered neuroanatomical changes such as hippocampal and whole brain atrophy or development of white matter hyperintensities that are hallmark signatures of cerebrovascular damage[4,5]. However, individual differences in vulnerability to specific CVMs are not understood through group-level investigations, hindering the generalizability of findings to individual patients. This limitation stems from several factors: (1) conventional sMRI measures are unable to distinguish between the different CVMs, a key concern since each CVM carries varying dementia risks; (2) the underlying neuropathological processes are highly variable, leading to a spectrum of sMRI presentations that are not fully captured by diagnostic labels; (3) group-level investigations on small, selective samples are often underpowered to study the complex interplay between comorbid conditions present in real-world patients. To detect cerebrovascular changes early and measure disease severity and progression, therefore, robust and generalizable in vivo imaging markers that can quantify a specific CVM's impact on an individual patient's sMRI are needed. Crucially, such markers could help determine the factors that influence an individual's vulnerability to CVM impacts, and potentially inform clinical trials in target selection and treatment measurement.

Quantifying brain health from neuroimaging data is feasible with machine-learning techniques that enable the mapping of multivariate sMRI measures into low-dimensional composite indices. For instance, the Spatial Patterns of Abnormalities for Recognition of Alzheimer's Disease (SPARE-AD) is an individualized index reflecting the presence and severity of Alzheimer's disease (AD)-like patterns of atrophy in the brain[6] and is predictive of future cognitive decline[7]. Fueled by the integration and harmonization of multi-cohort MRI datasets[8], these techniques yield interpretable and generalizable markers[7–10], facilitating patient-level evaluations of disease severity. The current study aims to determine (i) whether machine-learning techniques can detect and quantify subtle brain imaging signatures related to individual CVMs, and (ii) whether these signatures can be detected even in the presence of additional CVMs.

Here, we leverage the SPARE framework to investigate the neuroimaging signatures of specific CVMs in a cognitively asymptomatic population. Using a large, diverse cohort from 10 neuroimaging studies, we characterize sMRI signatures of hypertension, hyperlipidemia, smoking, obesity, and type 2 diabetes, and quantify their severity at the individual level. We show that the resulting markers, known collectively as SPARE-CVMs, are better at detecting CVM-related brain changes when compared to conventional sMRI markers, particularly in mid-life and early sub-clinical stages of the corresponding CVMs. Additionally, we validate the models on an external dataset, assess their robustness across demographic subgroups, and evaluate the impact of co-occurring CVMs on SPARE-CVM scores. SPARE-CVMs are associated with brain beta-amyloid status and cognitive performance, suggesting their potential to inform early risk detection and clinical decision-making. We demonstrate that SPARE-CVMs will capture a nuanced phenotypic spectrum of imaging signatures, providing a more granular understanding of the impact of CVM on brain health beyond simple diagnostic categories.

## Results

### Overview of SPARE-CVM modeling and individualized severity estimation

An overview of the study workflow is provided in Fig. 1. A total of 20,000 participants from 10 studies in the Imaging-based coordinate SysTem for AGing and NeurodeGenerative diseases (iSTAGING) dataset, for whom sMRI imaging measures were available, were used for training and validation of CVM signatures (Table 1, Supplementary Information S1). Participants were between 45 and 85 years of age (mean age (standard deviation, SD) = 64.1 (8) years, 54.5 % Female), and

had no known cognitive impairment as defined by study-specific criteria. An independent validation dataset of $N = 17,096$ (mean age (SD) = 65.4 (7.4) years, 53.4 % Female) participants from the UK-Biobank study who were added to the 2020 data release (UKBIOBANK v1.7) was used to validate model results.

Five separate support vector classification models described in the Methods section were trained to detect and quantify spatial sMRI patterns for each CVM— hypertension (HTN), hyperlipidemia (HL), smoking (SM), obesity (OB), and type 2 diabetes mellitus (T2D), to derive SPARE-HTN, SPARE-HL, SPARE-SM, SPARE-OB, and SPARE-T2D indices, respectively. CVM statuses were dichotomized as present (CVM+) or absent (CVM−) based on study-provided categorical responses and medication status where available, and augmented using traditional cut-offs applied to the continuous clinical measures (Supplementary Information S3). Clinical definitions for ground truth CVM status and sample sizes used in training are provided in Supplementary Table 3. The distribution of the ground truth CVM− and CVM+ labels for the five target conditions indicates a highly heterogeneous sample with co-occurring CVMs (Fig. 2). More than 30% of the samples had two or more co-occurring conditions (Supplementary Fig. 2). Our ML configuration achieved better performance compared to other commonly employed ML models (Supplementary Information S4, Supplementary Fig. 3) despite moderate area under the receiver operating characteristic curve (AUC) values for the training (and validation) datasets, which ranged between 0.64 (0.63) for SPARE-SM to 0.70 (0.72) for SPARE-OB (Supplementary Table 4). Three-dimensional projections of the resulting SPARE-CVM indices are shown in Fig. 2A–D. Supplementary Fig. 4 illustrates the heterogeneity of clinical profiles and neuroimaging signatures observed at the individual level. Greater expression of CVM severity is quantified as large positive values along the corresponding bar in the graph. The diverse magnitudes of SPARE-CVMs within this marker panel highlight their ability to detect subtle, spatially distributed sMRI patterns that are not easily discernible through visual inspection.

### SPARE-CVMs reveal distinct CVM-related spatial patterns on sMRI

Overall, higher SPARE-CVMs were associated with lower GM and WM volumes and higher WMH volumes, although the spatial patterns and strengths of correlation varied (Fig. 3 and Supplementary Fig. 5). While SPARE-SM was associated with global volume loss (blue colors), the other SPARE-CVMs were associated with more spatially specific patterns of volume differences. It is important to note that age was treated as a confounding variable for sMRI measures and SPARE-CVMs in multiple regression analyses. Hence the positive association noted below may be interpreted not as increasing volume, but rather as relatively preserved at older ages, suggesting potential resilience to atrophy in CVM+.

Cortical GM patterns—higher SPAREs for all CVMs were associated with a pattern of cortical atrophy in frontal GM regions including the anterior and posterior insula, the frontal and central opercular regions, and parts of the inferior frontal gyri, in parietal regions including the postcentral and supramarginal gyri, and temporal GM regions including the planum polare and planum temporale. In the frontal lobe, lower volumes in the middle frontal gyri, and the orbital gyri were associated with higher SPARE-HTN, SPARE-HL, and SPARE-SM, the subcallosal area with higher SPARE-HTN, SPARE-HL, and SPARE-OB, the posterior orbital gyri with higher SPARE-HTN and SPARE-T2D, and the supplementary motor cortex and the medial parts of the precentral and superior frontal gyri with SPARE-HL and SPARE-SM. In the parietal lobe, lower volumes in the angular gyrus were associated with higher SPARE-SM and SPARE-T2D. In the temporal lobe, lower volumes in the entorhinal area were associated with higher SPARE-HTN, SPARE-SM, SPARE-OB, and SPARE-T2D, and the superior temporal gyri with higher SPARE-SM and SPARE-T2D. In the occipital lobe, lower volumes in the lingual gyri were

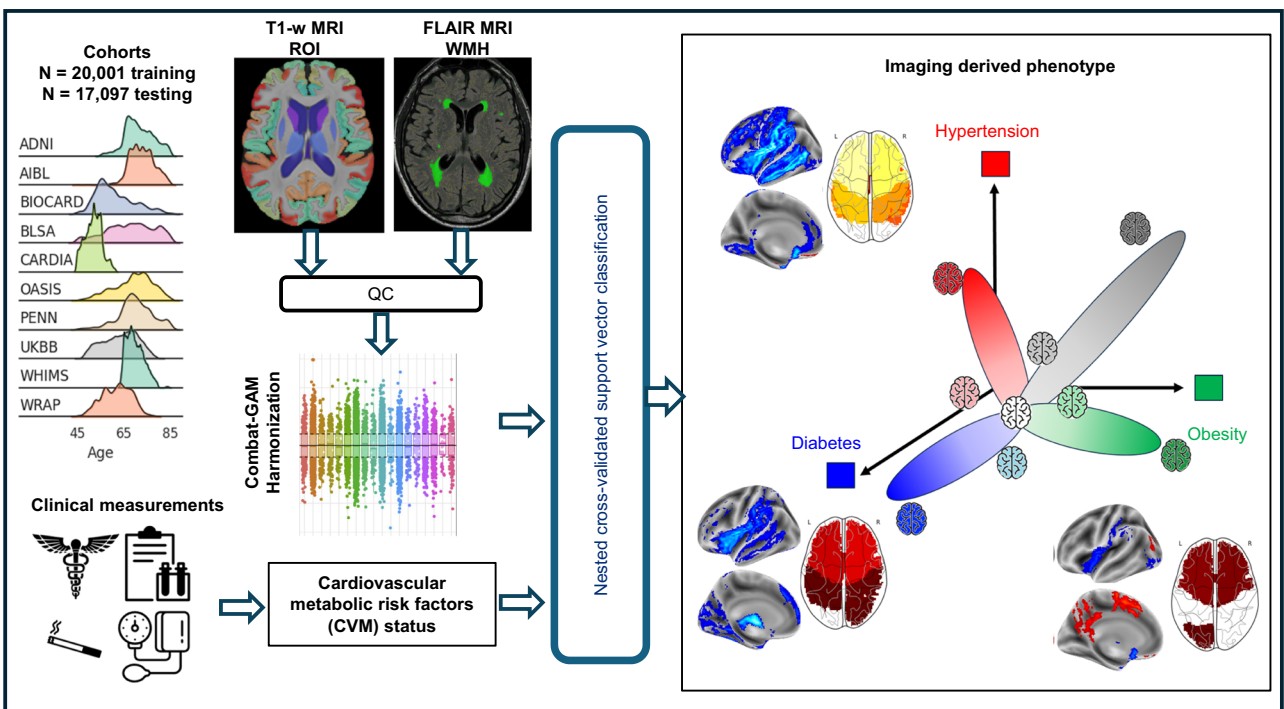

**Fig. 1 | Development of SPARE-CVM models.** Cardiovascular and metabolic risk factors (CVMs) contribute to distinct alterations in brain structure, potentially leading to diverse patterns of brain change. The current study leverages supervised machine learning (ML) to derive the phenotypic expression of CVMs from structural brain magnetic resonance images (sMRI). T1-weighted and T2-weighted FLAIR MRI from the multi-study iSTAGING dataset underwent rigorous preprocessing, segmentation, quality control (QC), and harmonization across study sites to obtain region of interest (ROI) brain volumes and white matter hyperintensity (WMH) volumes, respectively. These imaging features serve as inputs to the ML models. To establish ground truth labels for the ML models, clinical measurements, laboratory tests, and self-reported medical history were standardized to determine the presence or absence of five key CVMs: hypertension, hyperlipidemia, smoking, obesity, and type 2 diabetes mellitus. The ML models identify distinct Spatial Patterns of Abnormalities on sMRI associated with each CVM, summarized as individualized phenotypic expression scores (SPARE-CVMs), reflecting the influence of individual CVMs on brain structure.

**Table 1 | Overview of the data used for modeling imaging signatures of cardiovascular and metabolic risk factors (CVM)**

| Study | n | Years of MRI collection | Age (years) | Sex | Race (n) | Education years (n) | CVM prevalence (n CVM+) | | | | |
|---|---|---|---|---|---|---|---|---|---|---|---|
| | | | Median (range) | %F | White, Black, Asian, Other | <11, 11–14, >14 | HTN | HL | SM | OB | T2D |
| **Training dataset** | | | | | | | | | | | |
| Total | 20,000 | 1994–2020 | 65.1 (45–85) | 55 | 18,392, 815, 261, 23 | 11,353, 4896, 217 | 7865 | 6171 | 4018 | 4126 | 1511 |
| ADNI | 668 | 2006–2020 | 70.8 (55–85) | 60 | 340, 28, 5, 8 | 294, 84, 3 | 229 | 271 | 97 | 161 | 17 |
| AIBL | 569 | 2007–2019 | 72.6 (45–85) | 60 | 566, 0, 0, 0 | 200, 100, 192 | 261 | 199 | 139 | 96 | 43 |
| BIOCARD | 252 | 1999–2017 | 59 (45–85) | 61 | 245, 3, 3, 0 | 210, 37, 0 | 25 | 105 | 70 | 41 | 17 |
| BLSA | 934 | 1994–2019 | 68 (45–85) | 54 | 647, 226, 39, 20 | 766, 156, 6 | 390 | 115 | – | 217 | 138 |
| CARDIA | 829 | 2010–2016 | 52 (45–61) | 53 | 497, 331, 0, 0 | 481, 319, 28 | 278 | 298 | 134 | 308 | 182 |
| OASIS | 439 | 2000–2019 | 69.7 (46–85) | 58 | 372, 61, 3, 0 | 313, 120, 3 | 159 | 153 | 91 | 120 | 34 |
| PENN | 191 | 2010–2020 | 70 (46–85) | 67 | 141, 47, 0, 2 | 147, 39, 3 | 56 | 62 | 43 | 33 | 15 |
| UKBB v1.6 | 14810 | 2014–2019 | 64.1 (45–81) | 50 | 14,363, 71, 195, 17 | 8388, 3323, 1894 | 6096 | 4754 | 3296 | 2722 | 1000 |
| WHIMS | 1061 | 2004–2010 | 69 (64–84) | 100 | 976, 43, 15, 23 | 377, 639, 41 | 324 | 121 | 148 | 344 | 55 |
| WRAP | 257 | 2000–2016 | 62.5 (45–78) | 71 | 245, 5, 1, 5 | 177, 79, 0 | 47 | 93 | - | 84 | 10 |
| **Validation dataset** | | | | | | | | | | | |
| UKBB v1.7 | 17096 | 2017–2019 | 65.8 (48–82) | 53 | 16,495, 126, 243, 227 | 10,449, 3659, 185 | 7272 | 4418 | 3429 | 2867 | 892 |

Study descriptions are provided in Supplementary Information S1. Race, education, and CVM data were unavailable for some subjects (Supplementary Table 3).

associated with higher SPARE-HTN, SPARE-SM, and SPARE-T2D, and lower volumes in the cuneus and calcarine cortices were associated with higher SPARE-T2D. Relatively higher volumes in the middle occipital gyri and the cingulate gyri were associated with higher SPARE-HL and SPARE-OB, the gyrus rectus with SPARE-HL, the supplementary motor cortex, the precuneus and the occipital gyri with higher SPARE-OB.

Deep GM patterns—among deep GM structures, lower volumes in the accumbens area were associated with higher SPARE scores for all CVMs except SPARE-HL, the thalamus with higher SPARE-HL, SPARE-SM, and SPARE-T2D, and the pallidum associated with SPARE-SM, SPARE-OB, and SPARE-T2D. Relatively higher volumes in the hippocampus were associated with higher SPARE-HL and SPARE-OB, the putamen with SPARE-HTN and SPARE-HL, and the caudate nuclei with SPARE-HTN.

WM patterns—lower volumes of WM regions were associated with SPARE-HL, SPARE-SM, SPARE-OB, and SPARE-T2D. The strongest

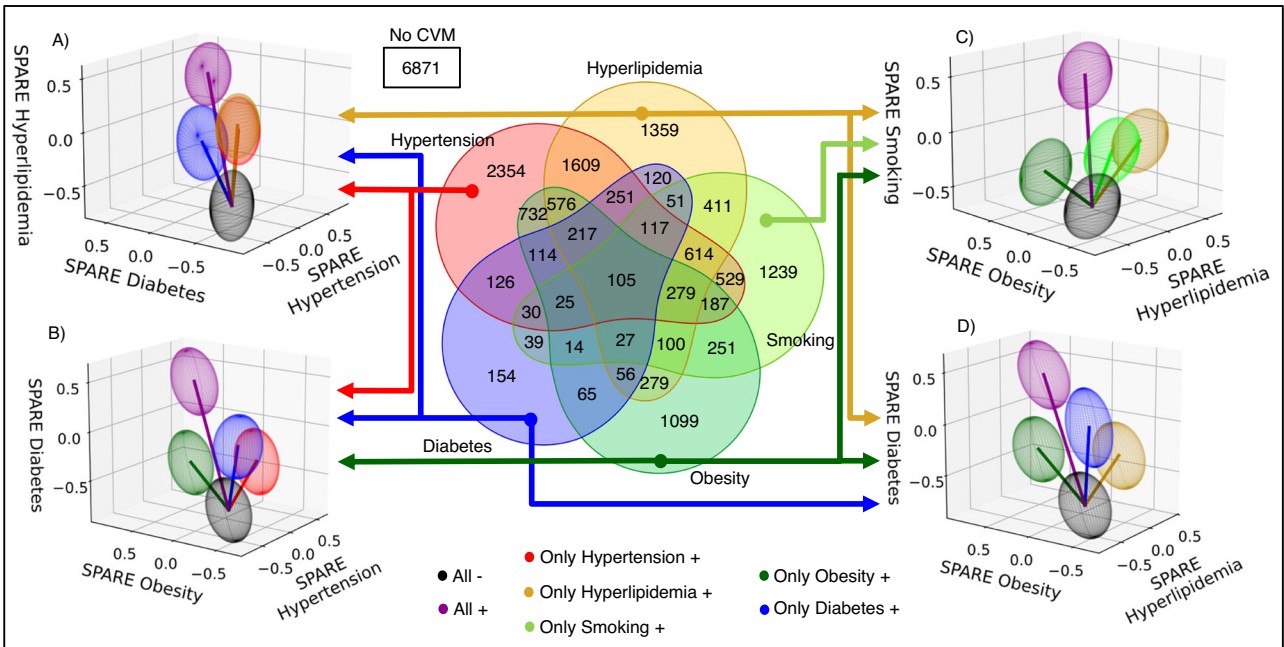

**Fig. 2 | CVM co-occurrence and multi-morbidity influence SPARE-CVM profiles across phenotypic dimensions.** The co-occurrence of cardiovascular and metabolic risk factors (CVMs) is prevalent in the general population, contributing to heterogeneity among participants and impacting the spatial patterns of abnormality related to CVMs (SPARE-CVMs). Center: A Venn diagram illustrates the CVM co-occurrence patterns observed in the training dataset. To visualize the influence of single- and multi-morbidity, three-dimensional projections of SPARE-CVMs are presented (**A–D**). Each ellipsoid within these projections represents the SPARE-CVM scores closest to the mean for a specific combination of CVM statuses. Participants with CVM− status for all three CVMs (All−) exhibit the lowest SPARE-CVMs, while those with CVM+ status for all three CVMs (All+) demonstrate the highest SPARE-CVMs. Notably, participants with CVM+ status in only one of the three CVMs show elevated scores specifically in the corresponding SPARE-CVM dimension, reflecting the distinct contributions of individual CVMs. Source data are provided as a Source Data file.

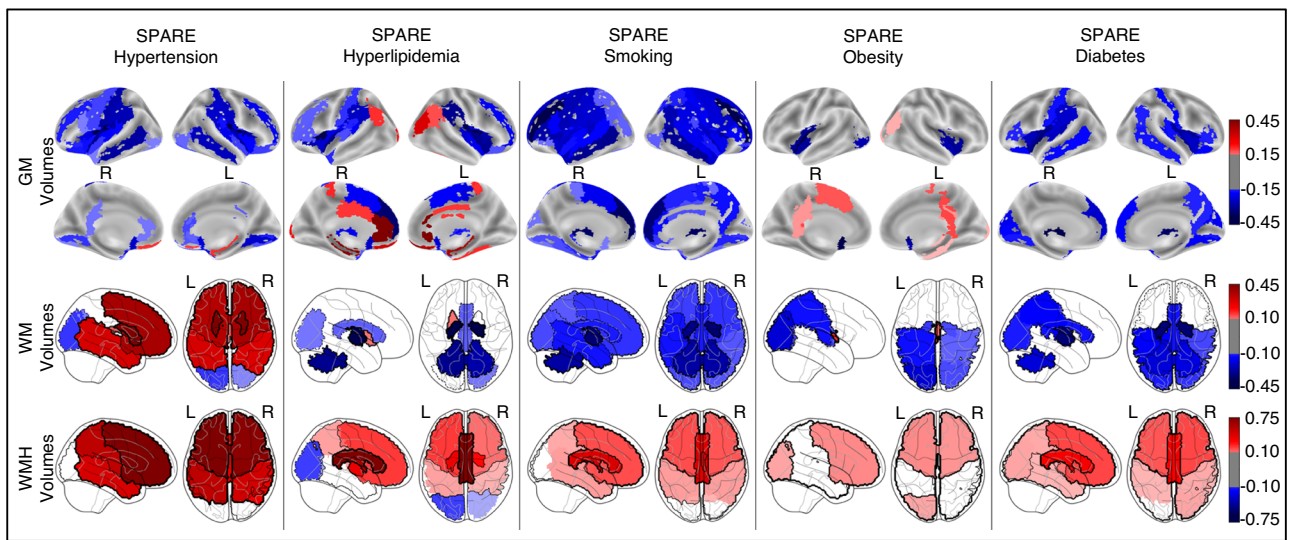

**Fig. 3 | SPARE-CVMs capture distinct spatial sMRI patterns of CVMs.** Volumes of gray matter (GM), white matter (WM), and white matter hyperintensities (WMH) showed significant associations with SPARE-CVMs ($p < 0.001$, Bonferroni corrected for multiple comparisons). Regional associations between GM volumes and SPARE-CVMs are visualized using 3D surface maps, displaying regression coefficients derived from two-sided multiple linear regression analyses. Lobar associations between WM and WMH volumes and SPARE-CVMs are visualized using glass brain plots, also displaying regression coefficients from two-sided multiple linear regression analyses. Hot colors (red) indicate positive associations (higher volumes associated with higher SPARE-CVMs), while cold colors (blue) indicate negative associations (lower volumes associated with higher SPARE-CVMs). SPARE Spatial Patterns of Abnormality Related to, GM Gray Matter, WMH White Matter Hyperintensities, L Left hemish.

associations were observed between volumes of the anterior internal capsule and cerebellar WM and SPARE-HL, SPARE-SM, and SPARE-T2D. In contrast, higher WM volumes were associated with higher SPARE-HTN. We speculate that this positive association is driven by the increased WMH presence which would contribute to the regional summary measures of WM extracted from T1-weighted sMRI.

WMH patterns—higher SPARE-HTN was associated with larger WMH volumes across most of the sub-cortical and deep WM partitions,

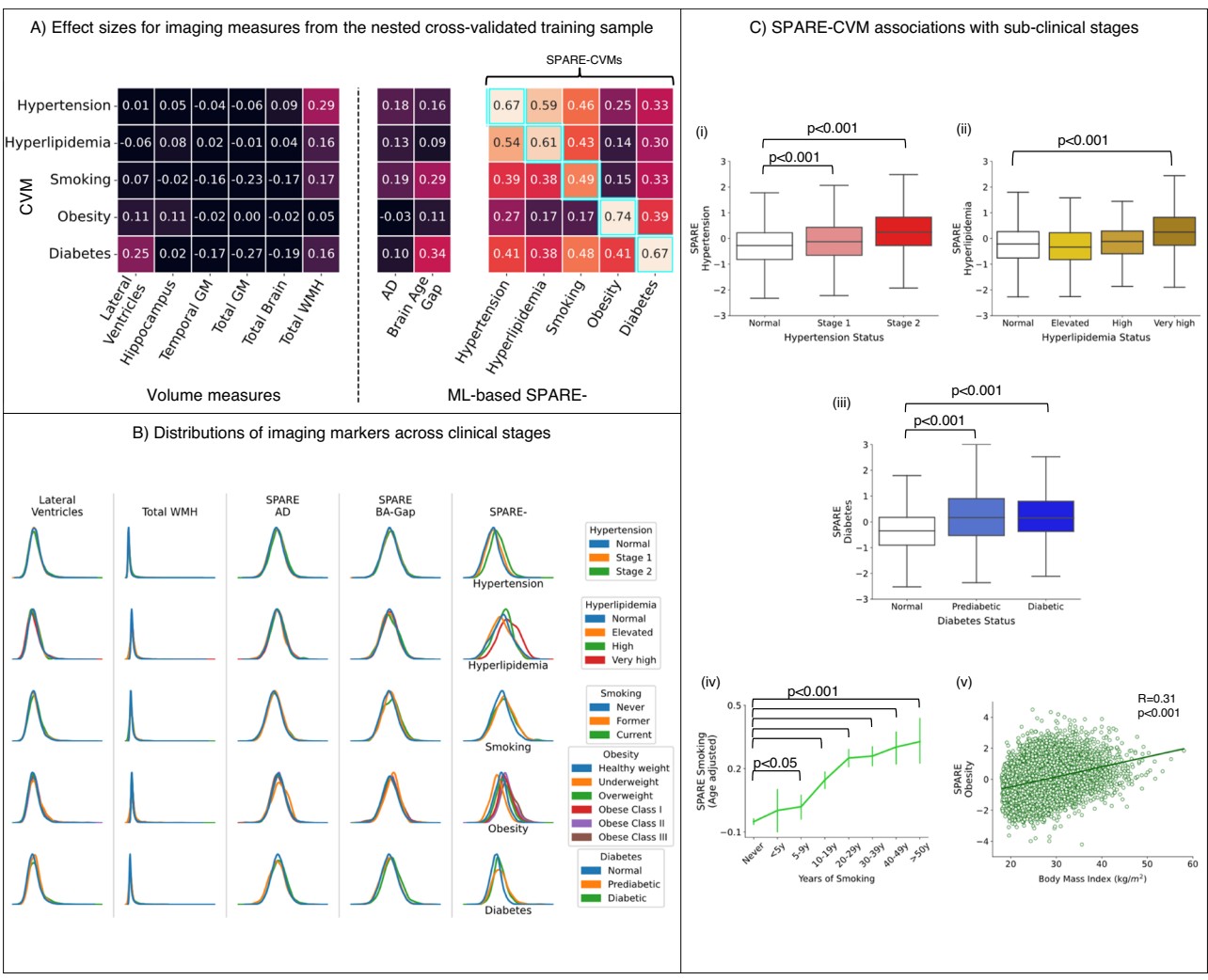

**Fig. 4 | SPARE-CVMs detect brain patterns more effectively across the clinical stages.** SPARE-CVMs exhibit enhanced sensitivity to CVM-related brain changes compared to conventional volumetric measures and machine-learning-based imaging markers for Alzheimer's disease and brain age. **A** A heatmap displays Cohen's $d$ effect sizes for each imaging marker (columns) at differentiating CVM+ and CVM− participants for the corresponding CVM (rows). The highest effect sizes for each CVM were observed in one-to-one correspondences between the SPARE model and the target CVM (outlined in blue). **B** SPARE-CVM indices show clear separability across clinical stages when compared to imaging markers which had small effect sizes in (**A**). Sub-clinical (undiagnosed) stages, defined by continuous clinical measures, were excluded from the training dataset (see "Methods"). **C** SPARE-CVM scores were significantly higher at sub-clinical stages, as demonstrated by: (i–iii)

Linear regression for categorical variables (with participants without CVM [CVM −/Normal] as the reference group) for SPARE-Hypertension, SPARE-Hyperlipidemia, and SPARE-Diabetes, visualized as boxplots (median and interquartile range). (iv) Linear regression for SPARE-Smoking (with participants who never smoked [Never/0 years] as the reference group), visualized as a line plot (mean ± 95% confidence interval). (v) Pearson's correlation for the correlation between SPARE-Obesity and body mass index (BMI), visualized in a scatterplot. All linear regressions were two-sided. Sample sizes, uncorrected $p$ values, and confidence intervals for the statistical tests are provided in the Source Data file. SPARE Spatial Patterns of Abnormality Related to, AD Alzheimer's Disease, CVM Cardiovascular and Metabolic Risk Factors, GM Gray Matter, WMH White Matter Hyperintensities.

but not with WMH in the occipital lobe. Higher WMH volumes in deep WM structures were associated with higher SPARE-HL, SPARE-SM, and SPARE-T2D.

**SPARE-CVMs are more sensitive to target CVMs than other imaging markers and are robust across demographic subgroups**
All 5 SPARE-CVMs showed medium-to-large Cohen's $d$ effect sizes in separating the corresponding CVM+ from CVM− individuals with $d = 0.67, 0.61, 0.49, 0.74$, and $0.67$ for HTN, HL, SM, OB, and T2D respectively. SPARE-CVMs showed the highest effect sizes for their target CVMs as seen along the diagonal elements in Fig. 4A outlined in blue. Replication on the independent dataset confirmed this finding (Supplementary Fig. 6A). Logistic regression analyses revealed that a unit increase in SPARE-CVM was associated with a higher odds ratio of having a positive status on the target CVM, ranging from OR = 1.79

(1.73–1.85) for SPARE-SM to OR = 2.39 (2.31–2.47) for SPARE-OB (Supplementary Fig. 6B). Additional sensitivity analyses revealed that SPARE-CVM effects are robust among male and female sexes (self-identified), self-identified race, and levels of education (Supplementary Fig. 7).

In contrast, effect sizes for univariate volumetric measures and ML-based SPARE-AD, SPARE-BA-Gap measures were largely inappreciable (Fig. 4A, Supplementary Fig. 8). Small effect sizes were observed for WMH volumes at separating HTN+/HTN− ($d = 0.29$) and ventricular volumes separating T2D+/T2D− ($d = 0.29$) individuals. SPARE-BA-Gap also showed small effect sizes with CVM+ participants having older brain ages than their chronological age $d = 0.16$ (+1.16 years) for HTN+, $d = 0.09$ (+0.6 years) for HL+, $d = 0.29$ (+2.1 years) for SM+, $d = 0.11$ (+0.9 years) for OB+, and $d = 0.34$ (+2.5 years) for T2D+ individuals.

### SPARE-CVMs capture brain changes in sub-clinical stages

Association between SPARE-CVMs and clinical stage or continuous measures of severity are shown in Fig. 4B, C. SPARE-HTN was significantly higher in participants categorized as "Stage 1" and "Stage 2" (+0.1 and +0.56, $p < <0.0001$) when compared to "Normal". SPARE-HL was significantly higher only in participants categorized as "Very high" (+0.52, $p < <0.0001$) when compared to "Normal", but not in "Elevated" or "High" categories. SPARE-T2D was significantly higher in participants categorized as "Prediabetic", with values on par with the Diabetic group (+0.69 and +0.61, $p < <0.0001$), when compared with "Normal". SPARE-HTN and SPARE-HL were elevated in medicated CVM+ individuals when compared to the unmedicated CVM+ individuals (SPARE-HTN + 0.28 and SPARE-HL + 0.52, $p < <0.0001$) but SPARE-T2D did not differ based on T2D medication status.

SPARE-OB correlated positively with BMI ($r = 0.31$, $p << 0.0001$). SPARE-SM was positively associated with years of SM after adjustment for age, increasing with each additional decade of smoking from +0.09 ($p < 5 \times 10^{-3}$) for 5–9 years to +0.42 ($p << 0.0001$) for >50 years of smoking.

### SPARE-CVMs are more pronounced in mid-life

Age associations of effect sizes showed that effect sizes peaked at the 45–50 years age interval for SPARE-T2D, the 50–55 interval for SPARE-HL and the 60–65 interval for SPARE-HTN and SPARE-OB, and tapered off in older ages (Fig. 5). The decline in SPARE-CVM effect sizes with age was also independently confirmed through sensitivity analyses by training multiple SPARE models at separate age ranges (Supplementary Information S4.2, Supplementary Fig. 9).

### Simultaneous presence of multiple CVMs is associated with co-expression of respective SPARE indices

Three-dimensional projections of SPARE-CVM distributions in Fig. 2A–D, and Supplementary Fig. 10 show the influence of single- vs multi-morbidity on the corresponding SPARE scores. SPARE-CVMs for CVM+ individuals without comorbid conditions showed a clear separation in the corresponding dimension. For example, SPARE-OB in only-OB+ cases showed separation from SPARE-HTN, SPARE-HL, and SPARE-SM dimensions (Fig. 2B–D). By contrast, SPARE-CVMs for commonly comorbid CVMs showed higher co-expression likely due to the shared brain patterns and the co-occurrence of CVMs in the population. HTN+ and HL+ were separable by the non-target SPARE-model with small-medium effect sizes (Fig. 4A) and SPARE-HTN and SPARE-HL scores overlapped in participants with only-HTN+ and only-HL+ (Fig. 2A). T2D was rarely present without comorbidities in our dataset, resulting in small effect sizes observed for non-target SPARE-CVMs. When the analysis was restricted to individuals with only one CVM when compared with individuals with no CVMs, the effect sizes remained highest for the SPARE models and target CVMs (Supplementary Fig. 10B). Additional logistic regression analyses with CVM status (+/−) as the outcome variable and all five SPARE-CVMs as predictor variables demonstrate higher specificity of SPARE-CVM to the target CVM but not the comorbid CVMs (Supplementary Fig. 10C).

### SPARE-CVMs had variable associations with amyloid deposition

A subset of our sample ($N = 407$) had amyloid status available within ±1 year of the MRI scan included in our dataset (Supplementary Table 5) and was included in multiple regression analyses to evaluate the interaction between amyloid deposition (Aβ+), CVM status and age on SPARE-CVM scores (Supplementary Fig. 11). Aβ+ participants were significantly older ($p < 0.001$) than Aβ- participants. Fewer participants were Aβ+ and OB+ when compared to Aβ+ and OB− in this cohort of CN participants ($p < 0.001$), perhaps suggesting that OB+ participants with Aβ+ were likely already experiencing cognitive symptoms warranting an MCI/dementia diagnosis or due to inclusion/exclusion criteria of the parent studies. SPARE-HTN in Aβ+ individuals was lower in

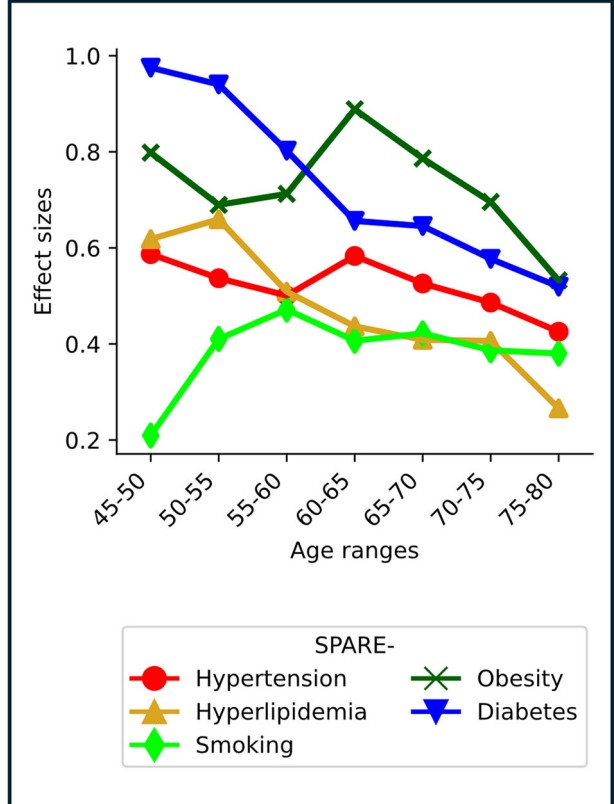

**Fig. 5 | SPARE-CVMs detect strong CVM effects in mid-life.** Peak effect sizes of SPARE-CVMs were observed in individuals aged 45–65 years, with a subsequent decline in effect sizes at older ages. Source data are provided as a Source Data file.

the younger ages of this sample (<70 years, $p < 0.01$) and increased with advancing age ($p < 0.05$), but did not reveal a significant interaction between HTN+ and Aβ+ status. SPARE-SM was lower in SM−Aβ+ individuals ($p < 0.05$) and higher in SM + Aβ+ individuals ($p < 0.05$), but did not show significant interactions between Aβ+ and age. Significant age interactions were observed between OB status and Aβ+, where SPARE-OB decreased with age in OB−Aβ+ individuals ($p < 0.05$) and trended towards an increase with age in OB + Aβ+ individuals ($p = 0.06$). No significant associations were found between Aβ+ and HL status or T2D status on the corresponding SPARE scores.

### SPARE-CVMs negatively correlated with cognitive performance

Significant negative associations ($p < 0.05$, corrected for multiple comparisons, Fig. 6) were observed between higher values for all SPARE-CVMs and lower DSST scores, longer time to complete TMT-A and TMT-B, and lower MOCA scores. Higher SPARE-HTN, SPARE-HL, SPARE-SM, and SPARE-T2D, but not SPARE-OB, were also associated with lower (by >10%) odds ratio of correct recall on the first attempt in the P-Mem test. Higher SPARE-HTN, SPARE-SM, SPARE-OB, and SPARE-T2D, but not SPARE-HL, were also associated with lower MMSE scores (Supplementary Fig. 12). In comparison, CVM status (+/−) was associated with performance on fewer cognitive tests.

### SPARE-CVMs from harmonized data were more generalizable across study sites

To evaluate the impact of the unwanted technical variations caused by site differences on SPARE-CVMs, we compared them with identical ML models trained with unharmonized MUSE ROI volumes as input features. We found that harmonization drastically reduced site-related variability in SPARE-CVMs, as evidenced by lower analysis of variance $F$ values (Supplementary Fig. 13). In contrast, SPARE scores trained using

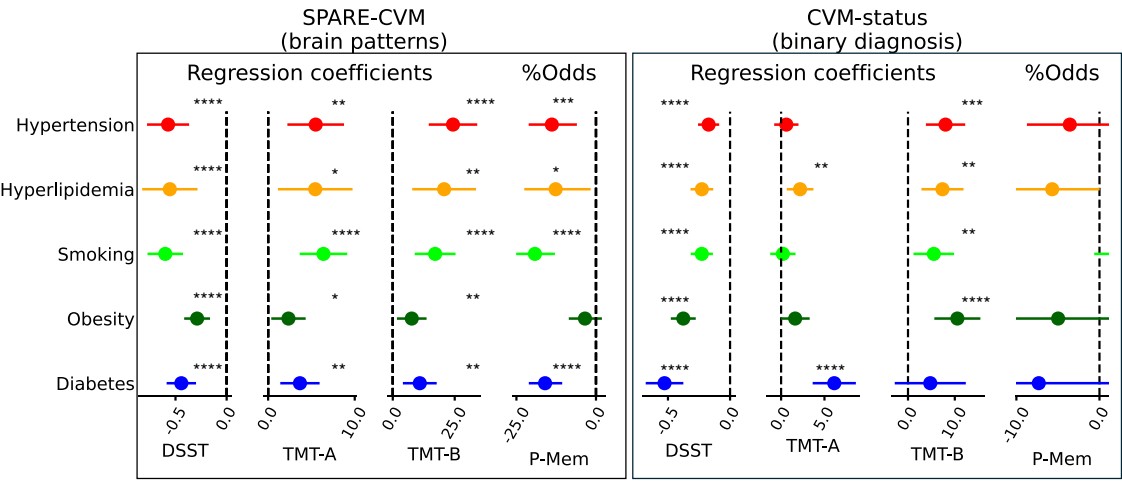

**Fig. 6 | SPARE-CVMs exhibit stronger associations with cognitive performance than CVM labels.** The results of the two-sided multivariate regression models predicting cognitive performance in UKBiobank using SPARE-CVMs (left) or CVM status (right), while adjusting for the confounding covariates of study, age, sex, and number of years of education, are shown below. Regression coefficients are represented by circles, with error bars indicating 95% confidence intervals. SPARE-CVM scores were adjusted for potential confounders, namely age, sex, and DLICV, prior to the regression analysis. Participants without CVM (CVM−) were the reference group in the regression models for diagnostic status. Odds ratio for smokers (SM+) for P-Mem is not shown on the CVM-status graph because it fell outside the axis limits (OR = 5.5, CI limits = −0.5,11.9, $p > 0.05$). For the Sample sizes, $p$ values, and confidence intervals are provided in the Source Data file. DSST Digit symbol substitution test, TMT−A/B Trail-making test −A/B; P-Mem Prospective memory. False discovery rate corrected $p$ values are indicated by: *$p < 0.05$, **$p < 0.01$, ***$p < 0.001$, and ****$p << 0.001$. Associations with $p$ values > 0.05 are not marked. For similar analyses on other cohorts in iSTAGING, please refer to Supplementary Fig. 12.

unharmonized MUSE ROI exhibited substantial site effects, limiting their generalizability beyond the original study sites despite effective CVM classification within those sites.

## Discussion

We leveraged machine-learning methods on a large multi-study sMRI dataset to tackle the persistent challenge of individual variability in CVM severity across the phenotypic spectrum in neuroimaging. ML-derived SPARE-CVM markers quantified characteristic neuroanatomical patterns associated with CVMs, detecting sub-clinical changes at the crucial period of mid-life ages, and showed strong associations with cognitive performance even in the absence of discernible cognitive impairments. Incorporating SPARE-CVMs into brain phenotyping can help assess the variable impact of CVMs on brain structural changes and their interactions with AD-related neurodegeneration at the individual level.

SPARE-CVMs showed better discrimination and more specific patterns of brain structure corresponding to each CVM, compared to established imaging markers including regional or whole brain volumes and other imaging markers such as SPARE-AD or SPARE-BA-Gap. In our models, we did not impose homogeneity or CVM exclusivity restrictions on our CVM+ participants, as evidence suggests that CVMs are more likely to occur together than separately at older ages and are commonly managed together in clinical practice[9]. Our results show that the shared expression of comorbid CVMs is captured by the individualized SPARE scores, enabling the investigations of the cumulative effects of comorbid CVMs.

The brain patterns associated with SPARE-CVMs are consistent with reported literature on CVM associations. Elevated lesion burden, particularly in the frontal WM, is a well-known indicator of vascular pathology. Similarly, the associations between elevated blood pressure frontal and temporal lobe atrophy[10], between smoking and total GM atrophy[7], between obesity and inferior frontal lobe atrophy, including the insular regions[11], and between diabetes and superior temporal gyri[12] have been reported in previous investigations. The positive association between GM volumes and CVMs suggests

relatively preserved volume at older ages, a result which appears unexpected but has nevertheless been previously reported in group-level investigations and meta-analytic studies[11,13,14]. Further multimodal imaging investigations are warranted to determine the possible mechanisms driving the positive association as it is unclear whether these associations were due to slower rates of age-related atrophy, or tissue hypertrophy driven by compensatory or inflammatory processes, or rather due to reduced myelin concentration exacerbated by CVMs, which can lead to poor contrast between sub-cortical GM and deep WM structures in sMRI. Importantly, these positive associations might also reflect a relatively higher brain reserve, especially in relatively unaffected brain regions, in older individuals who remain cognitively normal, despite the presence of CVMs.

SPARE-CVM scores showed associations with CVM severity. SPARE-HTN was higher in Stage 1 when compared to HTN-, SPARE-OB correlated BMI, and SPARE-SM increased with years of smoking suggesting a dose-dependent association between these CVMs and brain patterns. SPARE-T2D was higher in pre-diabetic and diabetic participants when compared to those with normal fasting glucose and HbA1c, confirming the association between poor glycemic regulation and brain alterations. Early brain changes have been observed in middle age[12] and older adults[15] in association with insulin resistance[16] and impaired glucose regulation resulting in high-normal glycemic levels[17], and even within a year of diabetes diagnosis[18].

We found greater separation in SPARE-CVMs between CVM− and CVM+ participants in mid-life ages when compared to older ages. This is likely to be driven by the multiplicity and heterogeneous progression of pathologic processes associated with aging and neurodegenerative diseases, confounding our ability to disentangle patterns of brain change specifically associated with individual CVMs at older ages. Conversely, relatively small brain changes expected at mid-life can stand out more vividly against a relatively lower background brain variability at this age range. Nevertheless, our models demonstrated the strongest effect sizes during the age range when the onset of CVMs carries the highest dementia risk. Age-varying associations of CVM onset with risk for cognitive decline and brain atrophy have been

reported, with elevated risk for mid-life onset of T2D[19], elevated blood pressure[20], and higher BMI[21], after adjustments for disease duration. SPARE-CVMs could thus have a profound impact in identifying individuals vulnerable to cerebrovascular changes at mid-life ages, a crucial time window for treatment and lifestyle interventions.

Our sensitivity analyses investigating the influence of amyloid deposition on SPARE-CVMs revealed interactive effects between Aβ+ and CVM+ status consistent with the literature. Individuals exhibiting elevated Aβ burden and vascular risk demonstrate more severe brain atrophy both cross-sectionally[22–24] and longitudinally[25–27] in regions vulnerable to AD, such as the parietal lobe and hippocampus, compared to individuals with only one risk factor. This was also observed in our whole brain summary indices, with higher SPARE-HTN, SPARE-SM, or SPARE-OB values in participants with HTN+, SM+, or OB + CVM status, respectively. CVMs occurring at mid-life appear to work synergistically with age, apolipoprotein E ε4, sex, and amyloid deposition to diminish brain integrity (atrophy, formation of white matter hyperintense lesions, and tau deposition)[24–26] and drive cognitive impairment (AD, vascular dementia). It is worth noting studies such as ADNI excluded participants with high vascular burden, and investigations on imaging and Aβ, including those reported here, may not capture a broad range of CVM risk. Further investigations with more population-representative datasets and stratified mediation analyses are needed to provide insights into the mechanistic pathways connecting CVMs, brain structural and functional alterations, amyloid pathology, and dementia.

Higher SPARE-CVMs were associated with lower cognitive test scores despite no overall group differences between CVM− and CVM+ participants, highlighting the value of analyzing the nuances of CVM impact on the brain beyond investigating clinical labels. This finding further emphasizes the potential of SPARE-CVMs in the early identification of individuals at risk of greater cognitive decline. Additionally, since these modifiable risk factors for dementia and cardiovascular disease converge[28–30], lifestyle interventions targeting blood pressure, lipid, and glucose control, along with reducing body mass and tobacco use in high-risk individuals, can potentially alleviate the burden of cardiovascular diseases.

Our study has limitations. We did not study participants with cognitive impairment, or follow-up AD diagnoses, and hence we did not evaluate whether SPARE-CVMs can predict future dementia. ML models were constructed to optimally separate CVM+ from CVM− individuals. CVM labels were derived from varying sources—self-reports, diagnosis codes, and health records, which were not uniformly available across all studies. However, to avoid training errors due to mislabeled false negatives, we used clinical measures where available and excluded participants with missing data. The cross-sectional nature of our study and the unavailability of information regarding disease or treatment duration across the sample pose a challenge in interpreting results related to medication status. SPARE-HL and SPARE-HTN were higher in individuals who received medical treatment for HL and HTN respectively, whereas SPARE-T2D was similar between treated and untreated T2D+ individuals. Further longitudinal investigation is needed as it is likely that these are not direct treatment effects, but rather arise from pharmacological intervention being more common for participants who presented with worse CVM symptoms and hence more pronounced brain differences. Our SPARE-CVMs show only moderate AUC levels around 0.7 for CVM classification. This study primarily aimed to develop quantitative markers for CVM-related sMRI patterns, rather than establish a diagnostic tool based on MRI. The expression or manifestation of these CVM signatures at the time of MRI acquisition can be highly variable across participants due to factors like differing disease duration, severity management, and individual susceptibility to brain and cognitive changes. By quantifying the CVM-related neuroanatomical effects, SPARE-CVMs are intended to help identify individuals with increased severity of brain alterations and not replace the far simpler and routine clinical diagnostic tools (e.g., blood pressure or weight measurements). For risk factors such as smoking, the severity of the impact on the brain may depend on additional factors such as the number of pack years and the time since smoking cessation. This could partially explain the low effect size observed for SPARE-SM, where CVM+ was determined based on the number of years of smoking to maximize available data across studies.

While this study has limitations, it benefits from a large multi-study dataset consisting of cohorts from diverse geographical and demographic backgrounds, and sMRI data harmonized using state-of-the-art techniques. SPARE-CVMs leverage sMRI, a commonly used neuroimaging tool in clinical settings, and can be applied for retrospective investigations of existing clinical datasets as well as secondary analyses of prior clinical trials such as the systolic blood pressure intervention trial (SPRINT) for hypertension treatment[31]. Being individualized measures, SPARE-CVMs can potentially inform and influence the design of future clinical trials targeting modifiable risk factors. In particular, trials targeting a particular CVM might be better powered to detect treatment effects by using the respective SPARE-CVM as the outcome measure, rather than using non-specific brain measures such as global or regional brain volumes. Moreover, such trials can use SPARE-CVMs for the selection and stratification of patients by risk for cerebrovascular changes. Furthermore, our integrated approach creates new opportunities to investigate the susceptibility to CVM-related neurodegeneration from non-modifiable risk factors such as sex, race, and genetic factors.

In conclusion, this study presents a novel approach to quantify CVM-related brain changes using machine-learning-derived MRI markers. Our findings demonstrate that these markers have greater discriminatory power than conventional imaging markers, particularly in middle-aged individuals, enriching the array of imaging biomarkers for characterizing neuroanatomical signatures. By providing a more nuanced understanding of the impact of CVM on brain health, these markers have the potential to inform personalized diagnostics and patient management, guide the design of future clinical trials through increased precision of treatment effect evaluations, and advance our understanding of the mechanisms underlying CVM-related neurodegeneration through enhanced detection of subtle CVM-related brain changes in longitudinal studies.

## Methods

### Study participants

The University of Pennsylvania institutional review board approved the protocols of this research study. We used a large multi-study collection of sMRI pooled and harmonized for the iSTAGING dataset. The studies in iSTAGING were carried out in diverse international geographic locations, with varying scanners and acquisition parameters between 1995 and 2020 (Supplementary Information S1). Participants provided written informed consent to the corresponding source studies. The analysis focused on cross-sectional scans for a subgroup of participants within the iSTAGING dataset, all of whom had complete sMRI measures available (Table 1).

### MRI preprocessing and harmonization

T1-weighted anatomical images were segmented into gray (GM) and white matter (WM) regions of interest using the multi-atlas, multi-warp segmentation (MUSE) tool[32], which uses an ensemble of diverse brain atlases and is relatively robust to MRI scanner and protocol variations. Regional volumes from the multi-site studies were harmonized using ComBat-GAM[8]. This technique corrects for systematic variations in brain volumes caused by differences in scanning equipment—such as the manufacturer and model of the scanner, the magnetic field strength, the hardware and sequences used—while ensuring that inherent biological differences among individuals, such as age, gender,

and brain size, are preserved. DeepMRSeg, a deep-learning-based segmentation tool, was used to derive total intracranial volumes (ICV) from T1-weighted images and WMH fluid-attenuated inversion recovery (FLAIR) and/or T2-weighted images[33]. WMH volumes were summarized within the lobar and deep WM regions. Raw images and segmentation masks underwent a previously established two-step semi-automated quality control procedure[34], available as a standalone software package MRISnapshot (https://cbica.github.io/MRISnapshot/). The procedure automatically ranked scans based on a quality score derived from segmented ROI volumes and flagged segmentations that deviated most from expected volume distributions. Flagged segmentations were visually inspected using visualization reports created by MRISnapshot to identify and exclude poor-quality segmentations. All regional volume measures of GM, WM, and WMH were adjusted for age, sex, and ICV, and the residuals were standardized to zero-mean and unit-variance for the ML models. See Supplementary Information S2 for a list of imaging features and an illustration of harmonization outcomes.

### Clinical data consolidation
Clinical measurements for CVM status were collected across studies using standard operating procedures—systolic and diastolic blood pressure for HTN, fasting blood measures for type 2 diabetes and hyperlipidemia, height and weight for estimating body mass index, and self-reports for smoking status. CVM statuses were dichotomized as present (CVM+) or absent (CVM−) based on study-provided categorical responses and medication status where available, and augmented using traditional cut-offs applied to the continuous clinical measures (Supplementary Information S3).

### Machine-learning models
Supervised classifiers were trained independently for each of the five CVMs to separate input features of CVM+ and CVM−, with no restrictions imposed for comorbidities. Linear support vector classifiers are well suited for our study goals. The decision boundary between classes are linear combinations of high-dimensional features, which makes them not only more interpretable but also reduces the risk of overfitting when compared to more complex decision boundaries. Age, sex, ICV, harmonized local GM and WM volumes, and lobar WMH volumes were used as input features ($n = 157$) using a nested cross-validation procedure to reduce overfitting (Supplementary Information S4.1)[35]. The training sets were stratified by CVM status for the nested k-fold cross-validation procedure to ensure an equivalent distribution of CVM− and CVM+ samples in each fold. Additionally, model performance was evaluated using scikit-learn's balanced accuracy scorer, which weights raw accuracy by the inverse of class prevalence, thereby making the models less susceptible to class imbalance issues. The classifiers output continuous scalar values that summarize the degree to which SPARE-CVMs are expressed in an individual's features. High/positive values suggest a clear presence and low/negative values suggest a relative absence of CVM-related patterns in the brain.

### Dissecting SPARE-CVM spatial patterns
To understand which imaging features most contributed to the SPARE-CVMs, associations between SPARE-CVMs and ROI/WMH volumes were assessed using multiple linear regressions for each sMRI feature adjusting for age, sex, and ICV. The resulting $p$ values were corrected for multiple comparisons using the Bonferroni method using an alpha of 0.001.

### Evaluating the separability and sensitivity of SPARE-CVMs
Cohen's $d$ effect sizes for SPARE-CVM differences between corresponding sets of CVM+ and CVM− groups were calculated. Higher $d$ values indicate greater separability between the CVM+ and CVM− groups, with empirical benchmarks for small ($d = 0.2$), medium ($d = 0.5$), and large ($d = 0.8$) effect sizes[36]. Similar analyses were performed on general neuroimaging measures, such as the volumes of the lateral ventricles, hippocampus, temporal GM, total GM, and total WMH, as well as previously established ML-based imaging markers for the brain age gap, defined as the difference between brain age estimated from sMRI and chronological age (SPARE-BA-Gap)[37], and Alzheimer's disease (SPARE-AD)[6]. Effect size analyses were replicated for all ML-based imaging markers in the independent dataset (UKBIOBANK v1.7). We performed logistic regression to assess how a unit increase in each SPARE-CVM is associated with the odds of having a positive status for the target CVM.

### Evaluating the associations between SPARE-CVMs and clinical measures of CV health
To assess the clinical relevance of CVM signatures beyond the dichotomized CVM categories, we investigated the association between each SPARE-CVM and the underlying clinical measures. SPARE-CVMs were estimated for participants with sub-diagnostic clinical measures who were excluded from the model training: "Stage 1" for HTN (systolic blood pressure: 130–150 mmHg OR Diastolic blood pressure: 80–95 mmHg), "Elevated" for HL (Triglycerides: 150–199 mg/dL OR LDL: 130–159 mg/dL), and "Prediabetes" for T2D (Fasting glucose 100–125 mg/dL OR HBA1C: 5.7–6.4 %). Additionally, SPARE-CVMs were compared between medicated and unmedicated CVM+ individuals with HTN, HL, and T2D. Pearson's correlation was used to evaluate the association between SPARE-OB and continuous measures of BMI, including overweight (BMI 25–30 kg/m$^2$) individuals who were not part of the model training. SPARE-SM was tested for its association with the number of years of smoking, a continuous measure that offers a partial estimate of risk severity.

### Evaluating the associations between SPARE-CVMs and markers of AD pathology
We performed additional sensitivity analyses to evaluate the interactions between age, CVM status, and the presence of AD pathology on SPARE-CVMs using multiple linear regressions. The presence of amyloid pathology (Aβ+) was determined using cerebrospinal fluid biomarkers (CSF) and mean cortical positron emission tomography standardized uptake ratios (SUVR). Study-specific cut-offs for CSF amyloid beta (Aβ42) concentrations were <192 pg/mL for ADNI[38] and <374.5 pg/mL for BIOCARD[39]; for Pittsburgh compound B ([11 C]PiB) SUVR ≥ 1.6 for ADNI[40], ≥1.5 for AIBL[41], ≥1.06 for BLSA[42], and >1.5 for OASIS[43]; for Florbetapir ([18 F]AV-45) SUVR ≥ 1.11 in ADNI and OASIS[44,45].

### Evaluation of associations between SPARE-CVMs and cognitive performance
We examined the association between SPARE-CVMs and cognitive performance in the following tests within the UKBioBank cohort: the digit symbol substitution test (DSST) for processing speed, the trail-making test (TMT-A and TMT-B) for executive function, and the prospective memory test (P-Mem). Multivariate linear regression models (for continuous measures) and logistic regression models (for P-Mem) predicting cognitive performance using SPARE-CVMs, while adjusting for the confounding covariates of age, sex, and years of education, were implemented. The resulting $p$ values were corrected for multiple comparisons using the Benjamini−Hochberg procedure for false discovery rates. Similar models were constructed for other cognitive tests that were available for a subset of studies, with additional corrections for the study data origin as a confounding variable (Supplementary Information S5.2).

### Reporting summary
Further information on research design is available in the Nature Portfolio Reporting Summary linked[36] to this article.

## Data availability

SPARE-CVM indices derived in this study have been uploaded as Supplementary Data in the Source Data file. Original imaging and clinical data used in this study were obtained through data-sharing agreements from the following ten individual studies: Alzheimer's Disease Neuroimaging Initiative (ADNI), Australian Imaging, Biomarker and Lifestyle Flagship Study of Ageing (AIBL), Biomarkers of Cognitive Decline Among Normal Individuals (BIOCARD), Baltimore Longitudinal Study of Aging (BLSA), Coronary Artery Risk Development in Young Adults (CARDIA), Open Access Series of Imaging Studies (OASIS), Penn Memory Center (PENN), UK Biobank (UKBB), Women's Health Initiative Memory Study (WHIMS), and Wisconsin Registry for Alzheimer's Prevention (WRAP). The data-sharing agreements do not include permission for us to share the data further. Investigators must apply to the source data providers to access additional data and match their subject IDs to those used in this study under the current protocol (primarily for UKBB). Data from ADNI and AIBL are available from the Imaging and Data Archive database (https://ida.loni.usc.ed) upon registration and compliance with the data usage agreement. Data from the UKBB are available upon request from the UKBB website (https://www.ukbiobank.ac.uk/). Data from the BLSA study are available upon request at https://www.blsa.nih.gov/how-apply. Data from the OASIS study are available upon request at https://www.oasis-brains.org/. Data requests for BIOCARD, PENN, WRAP, CARDIA, and WHIMS datasets should be directed to M.S.A., D.A.W., S.C.J., L.J.L., and M.A.E., respectively. Upon obtaining access to the source data, investigators can match our derived SPARE-CVM indices to the rest of the data from these studies. Further assistance in matching the R-indices can be requested from the corresponding last author, C.D., at Christos.Davatzikos@pennmedicine.upenn.edu, with responses typically provided within 2 weeks. Moreover, we are actively following protocols to upload our derived measures to the UKBB and ADNI websites, making them directly accessible to investigators who obtain access to those studies. Source data are provided with this paper.

## Code availability

Modeling and analyses utilized Python (version 3.8.1) and the models were developed using scikit-learn (version 1.3.2). We used several other Python and R libraries to support data analysis and visualization, including pandas (version 1.5.3), statsmodels (version 0.13.2), numpy (version 1.22.4), matplotlib (version 3.5.13), seaborn (version 0.12.2), scipy (1.7.3), ggplot2 (version 3.4.4), and venn (version 1.11). Python scripts for data processing are available on GitHub: https://github.com/CBICA/NiChart. Machine-learning models used in this project are available on GitHub via Zenodo: https://doi.org/10.5281/zenodo.14872922. This study's SPARE-CVM models and normative distributions are available in the NiChart: Neuro Imaging Chart of AI-based Imaging Biomarkers platform (https://neuroimagingchart.com/). NiChart allows researchers across the globe to upload their study data, process structural MRI for deriving volumetric features, harmonize said features to iSTAGING dataset, and predict SPARE-CVMs. The visualization modules in NiChart will then allow comparisons of the derived imaging signatures with the normative data in the dimensional coordinate system or as distribution plots.

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

## Acknowledgements

The iSTAGING study is a multi-institutional effort funded by the National Institute on Aging (NIA) by RF1 AG054409 (C. Davatzikos). Data used in preparation of this article were obtained from the Alzheimer's Disease Neuroimaging Initiative (ADNI) database (adni.loni.usc.edu). As such, the investigators within the ADNI contributed to the design and implementation of ADNI and/or provided data but did not participate in the analysis or writing of this report. A complete listing of ADNI investigators can be found at: http://adni.loni.usc.edu/wp-content/uploads/how_to_apply/ADNI_Acknowledgement_List.pdf. ADNI is funded by the NIA, the National Institute of Biomedical Imaging and Bioengineering, and through generous contributions from the following: AbbVie, Alzheimer's Association; Alzheimer's Drug Discovery Foundation; Araclon Biotech; BioClinica; Biogen; Bristol-Myers Squibb; CereSpir; Cogstate; Eisai; Elan Pharmaceuticals; Eli Lilly and Company; EuroImmun; F. Hoffmann-La Roche and its affiliated company Genentech; Fujirebio; GE Healthcare; IXICO; Janssen Alzheimer Immunotherapy Research & Development; Johnson & Johnson Pharmaceutical Research & Development; Lumosity; Lundbeck; Merck & Co; Meso Scale Diagnostics; NeuroRx Research; Neurotrack Technologies; Novartis Pharmaceuticals Corporation; Pfizer; Piramal Imaging; Servier; Takeda Pharmaceutical Company; and Transition Therapeutics. The Canadian Institutes of Health Research is providing funds to support ADNI clinical sites in Canada. Private sector contributions are facilitated by the Foundation for the National Institutes of Health (www.fnih.org). The grantee organization is the Northern California Institute for Research and Education, and the study is coordinated by the Alzheimer's Therapeutic Research Institute at the University of Southern California. ADNI data are disseminated by the Laboratory for Neuro Imaging at the University of Southern California. Data used in the preparation of this article were obtained from the Australian Imaging Biomarkers and Lifestyle flagship study of ageing (AIBL) funded by the Commonwealth Scientific and Industrial Research Organisation (CSIRO) which was made available at the ADNI database (www.loni.usc.edu/ADNI). The AIBL researchers contributed data but did not participate in the analysis or writing of this report. AIBL researchers are listed at www.aibl.csiro.au. The BIOCARD study is partly supported by NIH grant U19-AG033655 (M.S.A.). The BLSA neuroimaging study is funded by the Intramural Research Program, NIA, National Institutes of Health (NIH), and by HHSN271201600059C (S.M.R., M.B., Y.A.). CARDIA study is conducted and supported by the NHLBI in collaboration with the University of Alabama at Birmingham (HHSN268201300025C and HHSN268201300026C), Northwestern University (HHSN268201300027C), University of Minnesota (HHSN268201300028C), Kaiser Foundation Research Institute (HHSN268201300029C), and Johns Hopkins University School of Medicine (HHSN268200900041C). CARDIA is also partially supported by the Intramural Research Program of the National Institute on Aging (NIA) and an intra-agency agreement between NIA and NHLBI (AG0005) (L.J.L.). Data used in the preparation of this article was obtained from the OASIS study funded in part by grants P50 AG05681, P01 AG03991, P01 AG026276, R01 AG021910, P20 MH071616, U24 RR021382 for OASIS-1, P50 AG05681, P01 AG03991, P01 AG026276, R01 AG021910, P20 MH071616, U24 RR021382 for OASIS-2, and NIH P30 AG066444, P50 AG00561, P30 NS09857781, P01 AG026276, P01 AG003991, R01 AG043434, UL1 TR000448, R01 EB009352 for OASIS-3 (T.B., D.M., J.M., P.L.). Data used in the preparation of this article was obtained at Penn Alzheimer's Disease Research Center funded in part by grant P30 AG072979 (D.A.W.). Data used in the preparation of this article was obtained from the UK Biobank Resource under application number 35148. The Women's Health Initiative was funded by the National Heart, Lung and Blood Institute of the NIH, US Department of Health and Human Services. Contracts HHSN268200464221C and N01-WH-4-4221 provided additional support. The WHIMS (M.A.E.) was funded in part by Wyeth Pharmaceuticals. The WRAP study was supported by grants: NIH R01AG027161 and R01AG054047 (S.C.J.). The authors would like to acknowledge the clinical and neuropathology diagnostic support provided by the Wisconsin ADRC's Clinical, Neuropathology and Biomarkers Cores, and biostatistical support

provided by the Data Management and Biostatistics Core. S.T. Govindarajan was partly supported by the Alzheimer's Association Research Fellowship AARFD-23-1151286. A.A. was funded through grants 191026 and 206795 awarded by the Swiss National Science Foundation. M.H. was supported by grant 1R01AG080821 from the National Institutes of Health. Funding sources had no role in the study design, data collection, analysis, interpretation, or writing of the study report. The opinions and conclusions contained in this publication are solely those of the authors and are not necessarily endorsed by the associated studies, institutions, and funding agencies and should not be assumed to reflect their opinions or conclusions.

## Author contributions

Study concept and design was by S.T.G. and C.D. Model development was by S.T.G. Data interpretation was by S.T.G., H.S., I.M.N., and C.D. Drafting of the manuscript was by S.T.G., I.M.N., and C.D. Statistical analysis was by S.T.G. Data collection and processing was by S.T.G., E.M., G.E., A.A., R.M., J.D., R.P., D.T., M.B., Y.A., A.S., D.S.M., P.L., T.L.S.B., M.A.E., C.L.M., P.M., L.J.L., J.F., S.C.J., J.C.M., M.S.A., R.N.B., S.M.R., D.A.W., and C.D. Critical revision of the manuscript for important intellectual content was by S.T.G., E.M., G.E., A.A., R.M., J.D., R.P., D.T., M.B., Y.A., A.S., D.S.M., P.L., T.L.S.B., M.A.E., C.L.M., P.M., L.J.L., J.F., S.C.J., J.C.M., M.S.A., R.N.B., S.M.R., M.H., H.S., D.A.W., I.M.N., and C.D.

## Competing interests

T.L.S.B. has received investigator-initiated research funding from the NIH, the Alzheimer's Association, the Foundation at Barnes-Jewish Hospital, Siemens Healthineers, and Avid Radiopharmaceuticals (a wholly owned subsidiary of Eli Lilly and Company). She participates as a site investigator in clinical trials sponsored by Eli Lilly and Company, Biogen, Eisai, Jaansen, and Roche. She has served as a paid and unpaid consultant to Eisai, Siemens, Biogen, Janssen, and Bristol-Myers Squibb. J.C.M. has served as a paid consultant to the Barcelona Brain Research Center and the Native Alzheimer Disease-related Resource Center in Minority Aging Research. He also received payments for presentations at the AAIM meeting, Longer Life Foundation, and the International Brain Health Symposium. JCM has received travel support to attend meetings including AAIM, DIAN, AD/PD, ATRI/ADNI, ADRC, ADC, the International Conference on Health Aging & Biomarkers, and the International Brain Health Symposium. He has served on the advisory board for the Cure Alzheimer's Fund and LEADS at Indiana University. S.M.R. is an NIA IRP employee and has served on the advisory board of Dementia Platforms, UK, the Canadian Consortium on Neurodegeneration in Aging, and the Adult Aging Brain Connectome. She has received travel support from the McKnight Foundation to attend an annual meeting. D.A.W. has served as a paid consultant to Beckman Coulter and Eli Lilly. He also received grants from the NIH and Biogen paid to his institution and received travel support from the Alzheimer's Association. He has served on the DSMB of studies by Functional Neuromodulation and GSK. The other authors declare no competing interests.

## Additional information

[1]Center for Biomedical Image Computing and Analytics, University of Pennsylvania, Philadelphia, PA, USA. [2]Centre for Artificial Intelligence, ZHAW School of Engineering, Winterthur, Switzerland. [3]Department of Radiology and Biomedical Imaging, University of California, San Francisco, San Francisco, CA, USA. [4]Laboratory of Behavioral Neuroscience, National Institute on Aging, National Institutes of Health, Baltimore, MD, USA. [5]Department of Radiology, Washington University School of Medicine, St. Louis, MO, USA. [6]Sticht Center for Healthy Aging and Alzheimer's Prevention, Wake Forest School of Medicine, Winston-Salem, NC, USA. [7]Department of Biostatistics and Data Science, Wake Forest School of Medicine, Winston-Salem, NC, USA. [8]Florey Institute, The University of Melbourne, Parkville, VIC, Australia. [9]Neuroepidemiology Section, Intramural Research Program, National Institute on Aging, Bethesda, MD, USA. [10]CSIRO Health and Biosecurity, Australian e-Health Research Centre CSIRO, Brisbane, Queensland, Australia. [11]Wisconsin Alzheimer's Institute, University of Wisconsin School of Medicine and Public Health, Madison, WI, USA. [12]Knight Alzheimer Disease Research Center, Washington University in St. Louis, St. Louis, MO, USA. [13]Department of Neurology, Johns Hopkins University School of Medicine, Baltimore, MD, USA. [14]Department of Radiology, University of Pennsylvania, Philadelphia, PA, USA. [15]Biggs Alzheimer's Institute, University of Texas San Antonio Health Science Center, San Antonio, TX, USA. [16]Department of Biostatistics, Epidemiology and Informatics, University of Pennsylvania, Philadelphia, PA, USA. [17]Department of Neurology, University of Pennsylvania, Philadelphia, PA, USA. For the iSTAGING study, the Preclinical AD consortium, the ADNI, and the CARDIA studies: Christos Davatzikos.
✉e-mail: sindhuja.tirumalaigovindarajan@pennmedicine.upenn.edu; Christos.davatzikos@pennmedicine.upenn.edu

