## [Transparent Peer Review file · Nature Communications]

Machine learning reveals distinct neuroanatomical signatures of cardiovascular and metabolic diseases in cognitively unimpaired individuals

Corresponding Author: Dr Sindhuja Govindarajan

Version 0:

Reviewer comments:

Reviewer #1

(Remarks to the Author)

This is a very interesting study that evaluates, through machine learning, neuroanatomical signatures of cardiovascular and metabolic diseases in individuals with normal cognitive function. The results are very interesting for several reasons, especially for being able to harmonize the sMRI of several studies, for the large number of individuals evaluated and for the sound methodology.

I think that in general the article is very well written and clear from the point of view of the methodological description and results. Below I will make a few comments and suggestions that could improve the manuscript.

- 1) Even though the authors' objective is to evaluate the sMRI patterns of CVM disorders, it would be nice to see the interaction of all CVM variables with AD +, as these CVM are also risk factors to AD. Maybe as a sensitive analysis.
- 2) For a non specialist in machine learning methodology it is not so clear what the SPARE measure scale means. The authors explain in page 17 that "High/positive values suggest a clear presence, and low/negative values suggest a relative absence of CVM-related patterns in the brain." But for me it's not clear the difference between 0 to 1 and 1 to 2. Maybe the authors could explain it further.
- 3) Several charts are depicted in 3-D view. I don't think it is the best way to see the results, as sometimes it is hard to tell where each point is exactly. Maybe the authors could think in an alternative way to show some results.
- 4) In table 1: the data for the year when MRI were performed and age could also be seen as numeric data.
- 5) I don't think the figure 3 adds something to the comprehension of the results. Maybe it can be put as a supplementary table, as it serves as examples of individual findings.
- 6) In figure 6 it would be nice to see all the results, even the non significant ones.
- 7) In line 267: this sentence is unclear: "space could enable would help dissect CVM contributions to heterogeneous neuroanatomical".

(Remarks on code availability)

Reviewer #2

(Remarks to the Author)

In this paper, the authors tackled an important problem of identifying and understanding the associations between cardiovascular and metabolic risk factors and brain changes in a large cohort of patients. The topic is certainly worthy of investigation and very timely, thus interesting to the community. The manuscript, however, suffers from quite a number of shortcomings which should be, in my opinion, thoroughly addressed before the paper could be considered for publication:

1. The motivation behind selecting specific algorithmic components remains a bit obscured. As an example, the authors used support vector classification to derive the CVM models. Why the authors decided to use this model, given that there are lots of possibilities in the literature? What is the impact of this decision on the insights learned from this study? Also, did the authors consider performing appropriate training set selection for support vector classification engines, as it was shown to have a dramatic influence on the overall capabilities of this machine learning model?

2. The quality of the figures might be improved – some of them are of a rather low quality. All figures should be in a vector format, and they should be high-resolution.
3. The authors claim that the models were trained separately for all CVMs to “optimally” separate input features of CVM+ and CVM-. Can you formally prove this “optimality”?
4. As we are currently facing the reproducibility crisis in (not only) machine learning field, it is extremely important to ensure the reproducibility of the such studies. I appreciate seeing that the authors exploited a number of cohorts (which is indeed an advantage of this study), but it would be useful to provide a link to the implementation used in this work. The community should be able to reproduce the experiments reported in this work.
5. The authors may want to consider discussing the works that were aimed at determining the risk factors of e.g., cardiovascular diseases in patients with metabolic issues, in order to make the manuscript more self-contained.
6. Although the manuscript reads well overall, it would still benefit from careful proofreading – I spotted some minor grammar errors here and there around the manuscript.

(Remarks on code availability)

The authors should ensure reproducibility of their study (see my detailed comments).

Reviewer #3

(Remarks to the Author)

In this study, the authors developed and validated machine learning models to quantify the distinct spatial patterns of atrophy and white matter hyperintensities related to hypertension, hyperlipidemia, smoking, obesity and type-2 diabetes mellitus at the patient level using a large multinational dataset of 10 cohort studies. The authors demonstrated the advantages of the generated severity markers over conventional structural MRI markers and diagnostic CVM status in a few clinical tasks. Overall, the manuscript was interesting, well written and structured. However, some drawbacks on the novelty, methods and clinical applications, especially for the data analysis, were presented. Details of my concerns are as follows.

Major concerns:

1. In Methods-MRI preprocessing and harmonization: The author said that they used the ComBat-GAM to remove site effects on the MRI measures, while preserving nonlinear age-related differences in brain volumes. Some concerns were presented as follows: (1) It was not clear which confounding variables were input into the model. The author indicated the age was included the ComBat-GAM, how the non-linear associations between the age and brain volumes were assumed should be clearly demonstrated (e.g., quadratic or cubic curves); (2) As shown in Figure S-1, it seemed that the author only removed the site effects, however, Table S1-1 listed MR scanners with heterogeneous field strength (including both 1.5T and 3T), vendors (GE, Siemens and Philips) and protocols (MPRAGE and non-MPRAGE sequences). I wonder whether the author include all these site-related confounding factors into consideration in the ComBat-GAM, as the Figure S-1 just provided the harmonization results across sites. Additionally, some errors were observed in the Table S-1 regarding the protocols details, so I think the author may only take the site into consideration as the protocols they described were not correct (such as BIOCARD, if MPRAGE was applied, the TI was missing; ADNI, if MPRAGE was used for Philips, the TR and TI should not be same as Siemens, as the these vendors had different TR definitions for MPRAGE sequences; some protocols provided FOV, some did not; some provided slice number, some did not). Therefore, I have concerns on the MRI measure harmonization; (3) Another concern is that did the CVM was absent or only present in some of these sites, if so, the ComBat-GAM may reduce the sensitivity of the corresponding SPARE-CVM as the effects of the CVM may be reduced by the ComBat-GAM.
2. In Methods-MRI preprocessing and harmonization: Regarding on the brain tissue segmentation including T1-derived brain volumes and T2/FLAIR derived WMH. It seemed that the author did not conduct any quality controls neither on the raw data nor segmented volumes. Some segmentation errors may influence the individual findings especially for those with low quality of MR images. In addition, the author should provide how many cases had T1 images and how many had T2/FALIR images. Were these images all cross-sectional or some cases had longitudinal scans? Details should be provided to enhance the confidence of this study.
3. In Methods-Machine learning models: The authors had theoretically given the reasons why the linear support vector machine was used. However, additional comparisons with other machine learning methods may provide more information on the advantages of the linear SVM. A simple model is preferred for clinical utility. At least, the conventional logistic or lasso regression model should be compared to enhance the advantages of linear SVM.
4. In Methods-Machine learning models: The author demonstrated age, sex, ICV was also considered as input features in the machine learning model. However, they had said in the section of MRI preprocessing and harmonization, that all the regional volume measures of GM, WM and WMH were adjusted for age, sex and ICV, and the residuals were standardized. So, were the inclusion of age, sex and ICV necessary for the machine learning model? Even though the author included age in the machine learning, did a linear association (linear SVM model) assumption was suitable for the age. In addition, site effects on clinical variables may also present among these sites, was the site considered in the machine learning model? If not, the author should at least to explore whether there are some differences on the SPARE-CVMs among sites or centers (e.g., ADNI had multi-centers).
5. In Methods-Dissecting SPARE-CVM spatial patterns: The author assessed the associations between the calculated SPARE-CVMs and the ROI/WMH volumes using multiple linear regressions. Were all the ROI/WMH volumes considered in one multiple linear regression for each CVM, or was one ROI/WMH volume considered in one multiple linear regression for each MR feature and CVM?
6. In Results-SPARE-CVMs are more sensitive to target CVMs than other imaging markers and are robust across demographic subgroups: The author showed that the highest effect sizes were observed for their target CVMs. As the dataset is very large, did the author compare the findings using all populations and those using the “Normal” and cases with only one CVM (e.g., OB, SM and T2D as shown in Figure S-2). Additional comparisons may provide more evidences on

whether the SPARE-CVMs and related structural spatial patterns were CVM targeted. The other advice is that the author could use multiple linear regression with the five SPARE-CVMs as inputs and the CVM as outcomes, such analysis may assess the associations of the five SPARE-CVMs with CVM simultaneously.

7. In Results-Evaluation of associations between SPARE-CVMs and cognitive performance: Similar concerns were on the site effects on the cognitive scores, which should be considered in the regression models.

8. In Results, the performance (accuracy, sensitivity and specificity in training and validation sets) of the SVM for each CVM should be provided. As showed in the validation set, the AUCs seemed around 0.7 for these CVMs, indicating a relatively low classification ability of the SVM models. The SPARE-CVMs derived from SVM classification with low performance may reduce the credibility of the SPARE-CVM and its clinical utility.

9. The treatment effect and follow up of the individuals on the different SPARE-CVM is warrant to be investigated in further studies.

Minor concerns:

(1) The methods should be clearly described for a repeatability.

(2) A flow diagram regarding the data collection, processing and analysis etc. would make the study clear to readers.

(3) In Results, what's the exact meaning of $p < 0.0001$? Corrected it as $p < 0.0001$?

(4) The inclusion of brain age gap seemed weird. Please provide the detailed reason. Please also provide the reason for SPARE-AD.

(5) The contents in Table S-1 should be carefully modified and corrected.

(6) In Figure S-2, co-occurrence of CVM in the cross-validated training dataset should be also provided.

(Remarks on code availability)

Version 1:

Reviewer comments:

Reviewer #1

(Remarks to the Author)

I think the authors answer all the questions made by the reviewers appropriately. The changes made the paper clearer and straightforward and expanded some important subanalyses.

The results can contribute to the understanding of CV risk factors on dementia.

(Remarks on code availability)

The code provide by the authors is very detailed and contains a readme file with instructions for installing and running the code.

Reviewer #2

(Remarks to the Author)

Thank you indeed for addressing my concerns.

(Remarks on code availability)

Reviewer #3

(Remarks to the Author)

The authors have addressed most of my concerns, however, some details should still be clarified as follows.

Regarding comment 1:

“(2) As shown in Figure S-1, it seemed that the author only removed the site effects, however, Table S1-1 listed MR scanners with heterogeneous field strength (including both 1.5T and 3T), vendors (GE, Siemens and Philips) and protocols (MPRAGE and non-MPRAGE sequences). I wonder whether the author include all these site-related confounding factors into consideration in the ComBat-GAM, as the Figure S-1 just provided the harmonization results across sites.” The authors have provided examples that how they coded the “site” in the response. However, the absence of detailed information in the manuscript, supplementary materials, and the GitHub repository diminishes the credibility of the reported findings.

“Additionally, some errors were observed in the Table S-1 regarding the protocols details, so I think the author may only take the site into consideration as the protocols they described were not correct (such as BIOCARD, if MPRAGE was applied, the TI was missing; ADNI, if MPRAGE was used for Philips, the TR and TI should not be same as Siemens, as these vendors had different TR definitions for MPRAGE sequences; some protocols provided FOV, some did not; some provided slice number, some did not).” While the author has cited references that outline the protocols used, concerns remain unaddressed. Specifically, the OASIS study employs both 1.5T and 3T scanners, yet the protocols are identical, which seems implausible. Furthermore, the inversion time (TI) is set at a 20ms, and the definition of TD is unclear. Additionally, for

the UKBB dataset, the flip angle (FA) and echo time (TE) parameters are absent. It is crucial to ensure consistency in these protocols across different sites. A thorough check and clarification of these details are necessary

Regarding comment 8:

“8. In Results, the performance (accuracy, sensitivity and specificity in training and validation sets) of the SVM for each CVM should be provided. As showed in the validation set, the AUCs seemed around 0.7 for these CVMs, indicating a relatively low classification ability of the SVM models. The SPARE-CVMs derived from SVM classification with low performance may reduce the credibility of the SPARE-CVM and its clinical utility.” To facilitate a comprehensive evaluation of the models, it is essential to include the accuracy, sensitivity, and specificity metrics (preferably in the Supplementary materials). Although this study primarily aims to develop quantitative markers for CVM-related sMRI patterns, the performance of these models is paramount. This is because the reliability of the estimated quantitative markers is directly tied to the confidence in their predictive power.

(Remarks on code availability)

Version 2:

Reviewer comments:

Reviewer #3

(Remarks to the Author)

The authors replied my comments properly.

(Remarks on code availability)

We thank the reviewers for their time and feedback. We have addressed the questions (in blue) below and modified the manuscript as appropriate with tracked changes.

Reviewer #1 (Remarks to the Author):

1) Even though the authors' objective is to evaluate the sMRI patterns of CVM disorders, it would be nice to see the interaction of all CVM variables with AD +, as these CVM are also risk factors to AD. Maybe as a sensitive analysis.

To address the reviewer's comment, we performed additional sensitivity analyses which are now included in the manuscript. For the sake of clarity, we have chosen to use A β ⁺ to indicate the presence of AD pathology instead of AD+.

Methods:

Evaluating the associations between SPARE-CVMs and markers of AD pathology:

We performed additional sensitivity analyses to evaluate the interactions between age, CVM status, and the presence of AD pathology on SPARE-CVMs using multiple linear regressions. The presence of amyloid pathology (A β ⁺) was determined using cerebrospinal fluid biomarkers (CSF) and mean cortical positron emission tomography standardized uptake ratios (SUVR). Study-specific cut-offs for CSF amyloid beta (A β ₄₂) concentrations were <192pg/mL for ADNI [1] and <374.5pg/mL for BIOCARD [2]; for Pittsburgh compound B ([11C]PiB) SUVR \geq 1.6 for ADNI [3], \geq 1.5 for AIBL [4], \geq 1.06 for BLSA [5], and >1.5 for OASIS [6]; for Florbetapir ([18F]AV-45) SUVR \geq 1.11 in ADNI and OASIS [7, 8].

Results:

SPARE-CVMs had variable associations with amyloid deposition:

A subset of our sample (N=407) had amyloid status available within +/- 1 year of the MRI scan included in our dataset (**Table S-4**). A β ⁺ participants were significantly older (p<0.001) than A β ⁻ participants. Fewer participants were A β ⁺ and OB⁺ when compared to A β ⁺ and OB⁻ in this cohort of CN participants (p<0.001), perhaps suggesting that OB⁺ participants with A β ⁺ were likely already experiencing cognitive symptoms warranting an MCI/dementia diagnosis or due to inclusion/exclusion criteria of the parent studies. Results of the regression analyses are shown in **Figure S-11**). SPARE-HTN in A β ⁺ individuals was lower in the younger ages of this sample (<70 years, p<0.01) and increased with advancing age (p<0.05), but did not reveal a significant interaction between HTN⁺ and A β ⁺ status. SPARE-SM was lower in SM⁻ A β ⁺ individuals (p<0.05) and higher in SM⁺ A β ⁺ individuals (p<0.05), but did not show significant interactions between A β ⁺ and age. Significant age interactions were observed between OB status and A β ⁺, where SPARE-OB decreased with age in OB⁻ A β ⁺ individuals (p<0.05) and trended towards an increase with age in OB⁺ A β ⁺ individuals (p=0.06). No significant associations were found between A β ⁺ and HL status or T2D status on the corresponding SPARE scores.

Table S-4: Distribution of age, sex, and CVMs in participants with amyloid A β data.

Group differences between **A β -** and **A β +** were evaluated using two-sample t-tests for Age, and Chi-squared tests for categorical variables.

	A β -	A β +	p-value
N	292	115	
Age (SD)	69.3 (8.0)	73.3 (7.0)	p<0.001
Sex – Female N (%)	157 (53.8%)	63 (54.8%)	n.s.
CVM prevalence			
Hypertension N+/N-	118/115 (50.6 % HTN+)	38/48 (44.2% HTN+)	p=0.4
Hyperlipidemia N+/N-	107/112 (48.9 % HL+)	40/45 (47.1 % HL+)	p=0.9
Smoking N+/N-	55/107 (34.0 % SM+)	21/39 (35.0 % SM+)	p=1.0
Obesity N+/N-	90/63 (58.8 % OB+)	15/46 (24.6 % OB+)	p<0.001
Diabetes N+/N-	26/204 (11.3 % T2D+)	6/83 (6.7% T2D+)	p=0.3

Discussion

Our sensitivity analyses investigating the influence of amyloid deposition on SPARE-CVMs revealed interactive effects between A β and CVM status consistent with the literature. Individuals exhibiting elevated A β burden and vascular risk demonstrate more severe brain atrophy both cross-sectionally [9-11] and longitudinally [12-14] in regions vulnerable to Alzheimer's disease (AD), such as the parietal lobe and hippocampus, compared to individuals with only one risk factor. This was also observed in our whole brain summary indices, with higher SPARE-HTN, SPARE-SM, or SPARE-OB values in participants with HTN+, SM+, or OB+ CVM status, respectively. CVMs occurring at mid-life appear to work synergistically with age, apolipoprotein E ϵ 4, sex, and amyloid deposition to diminish brain integrity (atrophy, formation of white matter hyperintense lesions, and tau deposition) [24-26] and drive cognitive impairment (Alzheimer's disease, vascular dementia). It is worth noting studies such as ADNI excluded participants with high vascular burden, and investigations on imaging and A β , including those reported here, may not capture a broad range of CVM risk. Further investigations with more population-representative datasets and stratified mediation analyses are needed to provide insights into the mechanistic pathways connecting CVMs, brain structural and functional alterations, amyloid pathology, and dementia.

Figure S-11. Influence of A β status, CVM status, and Age on SPARE-CVM scores
The results of the multivariate regression models testing the interaction between age, CVM status and A β status in predicting SPARE-CVMs.

2) For a non specialist in machine learning methodology it is not so clear what the SPARE measure scale means. The authors explain in page 17 that "High/positive values suggest a clear presence, and low/negative values suggest a relative absence of CVM-related patterns in the brain." But for me it's not clear the difference between 0 to 1 and 1 to 2. Maybe the authors could explain it further.

SVMs model a hyperplane to maximize the margins between classes during training. The distance of the projection of the test data onto the vector perpendicular to the decision plane yields a continuous scalar. This is a unitless measure.

Figure 2 visually explains the interpretation of the SPARE-CVM scale (0 to 1 or higher). SPARE scores provide a single number to summarize complex patterns of brain changes associated with the target CVMs. The coefficient (β) values overlaid on the glass brain represent the volume differences observed along the corresponding pattern with a unit increase in SPARE-CVM scores.

In general, the higher the SPARE score, the more pronounced the corresponding imaging signature is in the sMRI.

3) Several charts are depicted in 3-D view. I don't think it is the best way to see the results, as sometimes it is hard to tell where each point is exactly. Maybe the authors could think in an alternative way to show some results.

Figure 3, which was formerly a 3-D plot has been moved to the supplementary file and the results section has been updated to include the following:

“Figure S-4 illustrates the heterogeneity of clinical profiles and neuroimaging signatures observed at the individual level. Greater expression of CVM-severity are quantified as large positive values along the corresponding bar in the graph. The diverse magnitudes of SPARE-CVMs within this marker panel highlight their ability to detect subtle, spatially distributed sMRI patterns that are not easily discernible through visual inspection.”

Figure S-4: Individualized SPARE-CVMs. Illustration of the heterogeneous clinical profiles, neuroimaging presentations, and individualized SPARE- scores from samples across 3 age ranges (top: 45-55, middle: 55-65, and bottom row: 65-75 years of age).

4) In table 1: the data for the year when MRI were performed and age could also be seen as numeric data.

We have updated Table 1 to reflect the age and years of MRI in numeric format. We included the median age to highlight that the age distributions may be skewed in individual studies.

Table 1: Overview of the data used for modeling imaging signatures of cardiovascular and metabolic risk factors (CVM). Study descriptions are provided in Appendix S1. Race, education, and CVM data were unavailable for some subjects (Table S-3).

Study	n	Years of MRI Collection	Age (Years)	Sex	Race (n)	Education Years (n)	CVM prevalence (n CVM+)				
			Median (Range)		%F	White, Black, Asian, Other	<11, 11-14, >14	HTN	HL	SM	OB
Training dataset											
Total	20001	1994-2020	65.1 (45-85)	55	18392, 815, 261, 23	11353, 4896, 217	7865	6171	4018	4126	1511
ADNI	668	2006-2020	70.8 (55-85)	60	340, 28, 5, 8	294, 84, 3	229	271	97	161	17
AIBL	569	2007-2019	72.6 (45-85)	60	566, 0, 0, 0	200, 100, 192	261	199	139	96	43
BIOCARD	252	1999-2017	59 (45-85)	61	245, 3, 3, 0	210, 37, 0	25	105	70	41	17
BLSA	934	1994-2019	68 (45-85)	54	647, 226, 39, 20	766, 156, 6	390	115	-	217	138
CARDIA	829	2010-2016	52 (45-61)	53	497, 331, 0, 0	481, 319, 28	278	298	134	308	182
OASIS	439	2000-2019	69.7 (46-85)	58	372, 61, 3, 0	313, 120, 3	159	153	91	120	34
PENN	191	2010-2020	70 (46-85)	67	141, 47, 0, 2	147, 39, 3	56	62	43	33	15
UKBB v1.6	14810	2014-2019	64.1 (45-81)	50	14363, 71, 195, 17	8388, 3323, 1894	6096	4754	3296	2722	1000
WHIMS	1061	2004-2010	69 (64-84)	100	976, 43, 15, 23	377, 639, 41	324	121	148	344	55
WRAP	257	2000-2016	62.5 (45-78)	71	245, 5, 1, 5	177, 79, 0	47	93	-	84	10
Validation dataset											
UKBB v1.7	17097	2017-2019	65.8 (48-82)	53	16495, 126, 243, 227	10449, 3659, 185	7272	4418	3429	2867	892

5) I don't think the figure 3 adds something to the comprehension of the results. Maybe it can be put as a supplementary table, as it serves as examples of individual findings.

Figure 3, which was formerly a 3-D plot has been updated and moved to the supplementary file (See response to R1/Q3 above).

6) In figure 6 it would be nice to see all the results, even the non significant ones.

Figure 6 for cognitive performance associations in the main manuscript and the corresponding one in the Appendix (Figure S-12) have both been updated to include all results. The non-significant results ($p > 0.05$) are left unmarked.

Figure 6: Association between cognitive performance and SPARE-CVMs or CVM status.

The results of the multivariate regression models predicting cognitive performance in UKBiobank using SPARE-CVMs or CVM status, while adjusting for the confounding covariates of study, age, sex, and number of years of education, are shown below. The scatter points (circles) in the figure represent the regression coefficients for each model. The lines show the limits of the 95% confidence interval for each coefficient. Odds ratio for smokers (SM+) for P-Mem is not shown on the CVM-status graph because it fell outside the axis limits (OR=5.5, CI limits= -0.5,11.9). CVMs were corrected for confounding covariates age, sex and DLICV. Abbreviations- DSST: Digit symbol substitution test, TMT-A/B: Trail making test -A/B; P-Mem: Prospective memory. False discovery rate corrected p-values are indicated by: * $p < 0.05$, ** $p < 0.01$, † $p < 0.001$, and †† $p < 0.0001$. Associations with p-values > 0.05 are not marked. For similar analyses on other cohorts in iSTAGING, please refer to Figure S-12.

Figure S-12: Association between cognitive performance and SPARE-CVMs or CVM status.

The results of the multivariate regression models predicting cognitive performance using SPARE-CVMs or CVM status, while adjusting for the confounding covariates of study, age, sex, and number of years of education, are shown below. The scatter points (circles) in the figure represent the regression coefficients for each model. The lines show the limits of the 95% confidence interval for each coefficient. SPARE-CVMs were corrected for confounding covariates age, sex and DLICV. Abbreviations- DSST: Digit symbol substitution test, MMSE: Mini-mental state examination; MOCA: Montreal cognitive assessment test; TMT-A/B: Trail making test A/B; MMSE: Mini-mental state examination; MOCA: Montreal cognitive assessment. False discovery rate corrected p-values are indicated by: ^ p = 0.05, * p < 0.05, ** p < 0.01, † p < 0.001, and †† p << 0.001. Associations with p-values > 0.05 are not marked.

7) In line 267: this sentence is unclear: “space could enable would help dissect CVM contributions to heterogeneous neuroanatomical”.

We updated the sentence for clarity.

“Incorporating SPARE-CVMs into brain phenotyping can help assess the variable impact of CVMs on brain structural changes and their interactions with AD-related neurodegeneration at the individual level.”

Reviewer #2 (Remarks to the Author):

1. The motivation behind selecting specific algorithmic components remains a bit obscured. As an example, the authors used support vector classification to derive the CVM models.

We have updated Appendix S4.1 section on ML models to reflect our rationale for choosing SVM in our study. Additionally, we included experimental validation of our choice in comparison to other common methods using AutoML (see additional details in response to R3/Q3 below), now included in Appendix S4.1

Why the authors decided to use this model, given that there are lots of possibilities in the literature? What is the impact of this decision on the insights learned from this study?

Using linear SVMs allowed us to model monotonic increases in SPARE-CVMs with direct relationships between feature weights and SPARE-CVMs, making them easier to interpret, while many other models, including deep learning, are less understandable. The trade-off from using linear models for SPARE-CVMs, as opposed to more complex non-linear modeling or deep learning methods, comes in the form of moderate AUC values of ~0.7. However, as we now demonstrate in Appendix S4.1, our models have superior performance to a host of other modeling techniques without running the risk of overfitting with more complex ML methods.

Also, did the authors consider performing appropriate training set selection for support vector classification engines, as it was shown to have a dramatic influence on the overall capabilities of this machine learning model?

We did perform appropriate training set selection for the SVM. We have updated the Methods section with information about the training set selection.

“The training sets were stratified by CVM status for the nested k-fold cross-validation procedure to ensure an equivalent distribution of CVM- and CVM+ samples in each fold. Additionally, model performance was evaluated using scikit-learn’s balanced accuracy scorer, which weights raw accuracy by the inverse of class prevalence, thereby making the models less susceptible to class imbalance issues.”

2. The quality of the figures might be improved – some of them are of a rather low quality. All figures should be in a vector format, and they should be high-resolution.

We thank the reviewer for this suggestion and have updated our figures for better quality.

3. The authors claim that the models were trained separately for all CVMs to “optimally” separate input features of CVM+ and CVM-. Can you formally prove this “optimality”?

We agree that the original wording was confusing. The term optimally was used to reflect the fact that SVMs find the maximum margin hyperplanes, i.e. the classifiers that maximally separate the positive from the negative examples. We have reworded the sentence as follows:

“Supervised classifiers were trained independently for each of the five CVMs to separate input features of CVM+ and CVM- , with no restrictions imposed for comorbidities.”

The details about the procedure and additional validation of model selection are presented in Appendix S4.1.

4. As we are currently facing the reproducibility crisis in (not only) machine learning field, it is extremely important to ensure the reproducibility of the such studies. I appreciate seeing that the authors exploited a number of cohorts (which is indeed an advantage of this study), but it would be useful to provide a link to the implementation used in this work. The community should be able to reproduce the experiments reported in this work.

We agree with the reviewer’s concerns about the reproducibility crisis. Data harmonized for this study were obtained from different sources with individual data usage agreements and cannot be shared with the manuscript. However, we are able to share the models on GitHub: https://github.com/stgovindarajan/spare_cvm_score. Centiles of expected values from the harmonized iSTAGING dataset are available through the NiChart:Neuro Imaging Chart of AI-based Imaging Biomarkers (<https://neuroimagingchart.com/>) platform. NiChart allows researchers to process their imaging data, derive the MUSE ROI segmentations used in the study, fit SPARE- models, and compare their results with the centile values from the reference set.

5. The authors may want to consider discussing the works that were aimed at determining the risk factors of e.g., cardiovascular diseases in patients with metabolic issues, in order to make the manuscript more self-contained.

We interpreted this comment as a suggestion to highlight that the risk factors investigated in our study are also part of the metabolic syndrome, an umbrella risk for cardiovascular disease. We have added the following to reiterate the common risk factors between cardiovascular diseases and dementia. We can make further revisions if we have misunderstood the reviewer's comment.

“Additionally, since these modifiable risk factors for dementia and cardiovascular disease converge [15-17], lifestyle interventions targeting blood pressure, lipid, and glucose control, along with reducing body mass and tobacco use in high-risk individuals, can potentially alleviate the burden of cardiovascular diseases.”

6. Although the manuscript reads well overall, it would still benefit from careful proofreading – I spotted some minor grammar errors here and there around the manuscript.

We thank the reviewer for the comment and revised our manuscript.

Reviewer #3 (Remarks to the Author):

1. In Methods-MRI preprocessing and harmonization: The author said that they used the ComBat-GAM to remove site effects on the MRI measures, while preserving nonlinear age-related differences in brain volumes. Some concerns were presented as follows:

(1) It was not clear which confounding variables were input into the model. The author indicated the age was included the ComBat-GAM, how the non-linear associations between the age and brain volumes were assumed should be clearly demonstrated (e.g., quadratic or cubic curves);

Linear terms for sex and total intracranial volume (ICV), and non-linear age term were adjusted in the ComBat-GAM model. Non-linear age was specified using a smooth spline term and estimated using penalized spline regression as in generalized additive model framework. The degree of smoothness was internally selected using the restricted maximum likelihood (REML) criterion.

For a demonstration and rationale for GAM modeling, we kindly refer the reviewer to Pomponio et al [18]. Briefly, Pomponio et al evaluated the goodness-of-fit for including age as linear, quadratic, and GAM (with a smoothed age term) models on several datasets spanning the life span (from adolescent to aging populations). Combat-GAM models achieved better performance in removing site differences than linear and quadratic models.

We edited the Methods section to reflect the confounding covariates used in the model:

“Regional volumes from the multi-site studies were harmonized using ComBat-GAM [18]. This technique corrects for systematic variations in brain volumes caused by differences in scanning equipment - such as the manufacturer and model of the scanner, the magnetic field strength, the hardware and sequences used - while ensuring that inherent biological differences among individuals, such as age, gender, and brain size, are preserved.”

(2) As shown in Figure S-1, it seemed that the author only removed the site effects, however, Table S1-1 listed MR scanners with heterogeneous field strength (including both 1.5T and 3T), vendors (GE, Siemens and Philips) and protocols (MPRAGE and non-MPRAGE sequences). I wonder whether the author include all these site-related confounding factors into consideration in the ComBat-GAM, as the Figure S-1 just provided the harmonization results across sites.

Although Figure S-1 shows harmonization effects by study, the harmonization model evaluates and removes effects at a much more granular level. We used ‘site’ as a general term for groups. Depending on the studies, the site variable was coded to include information about scanner features, such as manufacturer, field strength, and sequence differences, even though we did not directly adjust for scanner manufacturer, field strength and sequences as variables in the ComBat-GAM model. The “site” or batch for harmonization refers to the scanner site and parameter settings that stayed consistent during data collection. Studies like ADNI were grouped into ‘batch’ or ‘site’ based on their study phase collected (i.e., ADNI-1, ADNI-2/ADNI-GO,

ADNI-3) since the scans were prospectively standardized – i.e., the scanner field strength and sequences were adapted in a data-driven manner to ensure uniformity across the multicenter study [19]. In the case of studies like CARDIA, MRI data collection was performed across three different locations [20], each of which is treated as a separate ‘batch’ or ‘site’ in ComBat-GAM. Similarly, BLSA-1.5T [21] and BLSA-3T [22] are treated as separate sites in the harmonization model, as the scanning parameters within site/phase were maintained for longitudinal consistency [23].

Additionally, some errors were observed in the Table S-1 regarding the protocols details, so I think the author may only take the site into consideration as the protocols they described were not correct (such as BIOCARD, if MPRAGE was applied, the TI was missing; ADNI, if MPRAGE was used for Philips, the TR and TI should not be same as Siemens, as these vendors had different TR definitions for MPRAGE sequences; some protocols provided FOV, some did not; some provided slice number, some did not).

We have verified the data presented in Table S1 and made corrections where applicable. The scanner parameters reported in the supplementary file were received from the studies directly. We additionally verified these values with published work that used the corresponding studies. We included additional citations in the supplementary file, which detail the scanner protocol by site and study.

Therefore, I have concerns on the MRI measure harmonization;

To further clarify, Combat-GAM does not directly take scanner manufacturer, field strength and sequences as variables in the model. Rather we created ‘site’ variable by grouping data with consistent scanner site and parameter settings during data collection. Combat-GAM is a retrospective harmonization method used to mitigate scanner biases in the features derived from MRI (ROI volumes, in this study). Scanner-level differences in the manufacturer, magnetic field strength, head coil, gradient and receiving coils, sequence specification, voxel size, etc. are modeled as site-specific differences in mean and variance across the pooled ROI volumes. Corrective site-level shift and scale parameters are then applied to the data to ensure that the relationships between data points within and across sites/batches are maintained.

(3) Another concern is that did the CVM was absent or only present in some of these sites, if so, the ComBat-GAM may reduce the sensitivity of the corresponding SPARE-CVM as the effects of the CVM may be reduced by the ComBat-GAM.

Combat-GAM harmonization models were trained on the entire iSTAGING dataset which is substantially larger and contains more diverse cohorts than the subset used in this study. The model cohorts comprised cognitively unimpaired individuals from the general aging population, rather than clinically specific groups. None of the cohorts contained systematic biases or exclusion of CVM risk factors. Table 1 in the manuscript has been updated to also include the number of CVM+ participants by study in this subset of iSTAGING. Additional validation for building SPARE-CVMs using harmonized data is presented below in response to R3-Q4 below.

2. In Methods-MRI preprocessing and harmonization: Regarding on the brain tissue segmentation including T1-derived brain volumes and T2/FLAIR derived WMH. It seemed that the author did not conduct any quality controls neither on the raw data nor segmented volumes. Some segmentation errors may influence the individual findings especially for those with low quality of MR images.

Image processing leveraged established and validated methods implemented in fully automated pipelines to extract regional volumes of normal and abnormal tissues from MRI scans. Given the substantial sample size, a two-tiered QC process was implemented to ensure data quality. An automated ranking system, based on z-scores of segmented ROI volumes, initially prioritized scans by efficiently identifying outliers. Subsequently, both raw images and segmentation masks, guided by the automated ranking, were subjected to manual inspection using MRISnapshot, an in-house QC tool. MRISnapshot's user-friendly web interface allows for efficient navigation and tagging of large datasets by displaying overlays of selected labels directly on specific image slices in a configurable way (<https://github.com/CBICA/MRISnapshot>). The systematic QC methods were previously described in more details in a study by Srinivasan et al [24].

We updated the methods section of the manuscript to add a short description of the QC procedure and the above citation:

“Raw images and segmentation masks underwent a two-step semi-automated quality control procedure, as described previously in Srinivasan et al [24]. This procedure involved an automated ranking that assessed the distribution of segmented ROI volumes to identify those that deviated most from the expected ranges, followed by a manual review of raw and segmented scans, guided by the ranking value, using MRISnapshot, an in-house QC tool [25].”

In addition, the author should provide how many cases had T1 images and how many had T2/FALIR images. Were these images all cross-sectional or some cases had longitudinal scans? Details should be provided to enhance the confidence of this study.

All cases used in this study had T1 images and T2/FLAIR images available for deriving the sMRI features. We used cross-sectional images.

We included the following line in the Methods section for Study Participants.

“The analysis focused on cross-sectional scans for a subgroup of participants within the iSTAGING dataset, all of whom had complete sMRI measures available (**Table 1**).”

3. In Methods-Machine learning models: The authors had theoretically given the reasons why the linear support vector machine was used. However, additional comparisons with other machine learning methods may provide more information on the advantages of the linear SVM. A simple model is preferred for clinical utility. At least, the conventional logistic or lasso regression model should be compared to enhance the advantages of linear SVM.

To address the reviewer's comment regarding SVM selection, we conducted an automated machine learning (AutoML) experiment using the same dataset. We employed Auto-Sklearn [26], which leverages scikit-learn's algorithms. Auto-Sklearn explores different hyperparameter

configurations for various classification algorithms such as K-Nearest Neighbors, Random Forest Classifier, Linear Discriminant Analysis (lda), Quadratic Discriminant Analysis (qda), Multi-layer Perceptron (mlp), Passive Aggressive Classifier (passive_aggressive), Decision Tree Classifier (decision_tree), Extra Trees Classifier (extra_tree), Multinomial Naive Bayes (multinomial-nb), Gaussian Naive Bayes (gaussian_nb), Bernoulli Naive Bayes (bernoulli_nb) AdaBoost Classifier, Gradient Boosting Classifier, and Stochastic Gradient Descent Classifier (sgd).

As shown in the figures below, our initial SVM implementation achieved superior performance (balanced accuracy and AUC) compared to other models across all training folds. This finding validates our initial choice of SVM for this task.

We updated Appendix S4.1 to include these results:

Figure S-3: Comparison with other machine learning models.

Performance evaluation of our support vector classifier model configuration in comparison with other machine learning models. Metrics shown below are the area under the curve of the receiver operating characteristic (ROC AUC) and balanced accuracy evaluated on the “test” set across each outer fold of the nested cross-validation. Models compared across the outer folds: Support vector classifiers used in the study (K-Nearest Neighbors, Random Forest Classifier, Linear Discriminant Analysis (lda), Quadratic Discriminant Analysis (qda), Multi-layer Perceptron (mlp), Passive Aggressive Classifier (passive_aggressive), Decision Tree Classifier (decision_tree), Extra Trees Classifier (extra_tree), Multinomial Naive Bayes (multinomial-nb), Gaussian Naive Bayes (gaussian_nb), Bernoulli Naive Bayes (bernoulli_nb) AdaBoost Classifier, Gradient Boosting Classifier, and Stochastic Gradient Descent Classifier (sgd).

Hypertension

Hyperlipidemia

Smoking

Obesity

4. In Methods-Machine learning models: The author demonstrated age, sex, ICV was also considered as input features in the machine learning model. However, they had said in the section of MRI preprocessing and harmonization, that all the regional volume measures of GM, WM and WMH were adjusted for age, sex and ICV, and the residuals were standardized. So, were the inclusion of age, sex and ICV necessary for the machine learning model?

We would like to clarify that ComBat-GAM harmonization does not remove age, sex, and ICV effects. In fact, age, sex, and ICV were included as confounds in the harmonization models so that such biological variations could be preserved in the harmonized data and that the unwanted technical variations caused by site differences could be estimated and corrected with little bias. Hence, our harmonized data still contain age, sex, and ICV variations, which act as confounds in the models and need to be corrected. Further, the distribution of CVMs varies by Age and Sex, and including these variables in the model could capture subtle disease-related effects not addressed by confound correction on the entire cohort.

Even though the author included age in the machine learning, did a linear association (linear SVM model) assumption was suitable for the age.

Using linear SVMs allowed us to model monotonic increases in SPARE-CVMs with direct relationships between feature weights and SPARE-CVMs, making them easier to interpret. The trade-off from using linear models for SPARE-CVMs, as opposed to more complex non-linear modeling or deep learning methods, comes in the form of moderate AUC values of ~0.7. However, as we now demonstrate in Appendix S4.1, our models have superior performance to a host of other modeling techniques without running the risk of overfitting with more complex ML

methods. Future work with explainable non-linear models could enhance the effect sizes of SPARE-CVMs in older ages, where non-linear age trends may be more crucial.

In addition, site effects on clinical variables may also present among these sites, was the site considered in the machine learning model?

No, “site” was not included in the ML models. Our goal is to enable the prediction of SPARE-CVMs for sMRI from new studies or sites that the ML models were not previously trained on.

The clinical variables were consolidated across studies based on established clinical cutoffs, in addition to study provided binary diagnostic labels (Appendix S3). We also acknowledge that the clinical measures were not uniformly available across all cohorts in the Discussion section of the manuscript:

“CVM labels were derived from varying sources – self reports, diagnosis codes, and health records, which were not uniformly available across all studies. However, to avoid training errors due to mislabeled false negatives, we used clinical measures where available and excluded participants with missing data.”

If not, the author should at least to explore whether there are some differences on the SPARE-CVMs among sites or centers (e.g., ADNI had multi-centers).

To address this concern, we present a comparative evaluation of SPARE-CVMs trained using identical procedures, with unharmonized volumes as input features. MRI measurement differences between sites do propagate to SPARE-CVMs modeled using the original, unharmonized ROIs, making them less comparable across different studies. We have included the following results in Appendix S5.4.

We performed Analysis of Variance (ANOVA) to evaluate the impact of site differences on SPARE-CVMs trained using Harmonized and Unharmonized data. By comparing linear models with and without ‘site’ as a covariate, we found that harmonization drastically reduced site-related variability in SPARE-CVMs, as evidenced by lower F-values (Figure below). In contrast, Unharmonized SPARE-CVMs exhibited substantial site effects, limiting their generalizability beyond the original study sites despite effective CVM classification within those sites.

Figure S-13: Analysis of Variance (ANOVA) evaluation of site effects

F-values from ANOVA comparing the effect of ‘site’ in SPARE-CVMs trained using harmonized and unharmonized MUSE ROI volumes are shown below. Site effects have a substantial influence on SPARE-CVMs derived using unharmonized MUSE ROI volumes.

5. In Methods-Dissecting SPARE-CVM spatial patterns: The author assessed the associations between the calculated SPARE-CVMs and the ROI/WMH volumes using multiple linear regressions. Were all the ROI/WMH volumes considered in one multiple linear regression for each CVM, or was one ROI/WMH volume considered in one multiple linear regression for each MR feature and CVM?

Each SPARE-CVM and ROI/WMH volume were evaluated separately, adjusting for confounds. We clarify this in the methods:

“..associations between SPARE-CVMs and ROI/WMH volumes were assessed using multiple linear regressions for each sMRI feature adjusting for age, sex and ICV.”

6. In Results-SPARE-CVMs are more sensitive to target CVMs than other imaging markers and are robust across demographic subgroups: The author showed that the highest effect sizes were observed for their target CVMs.

As the dataset is very large, did the author compare the findings using all populations and those using the “Normal” and cases with only one CVM (e.g., OB, SM and T2D as shown in Figure S-2).

Yes. Figure 4A provides the effect sizes when comparing SPARE-CVMs between the ‘Normal’ group (absence of all CVMs) and individuals with target CVM+ without excluding comorbidities. Figure S-10B shows the same comparison between the ‘Normal’ group and groups with only the target CVM present. That is, participants with more than one CVM were excluded when calculating effect sizes in Figure S-10B.

Additional comparisons may provide more evidences on whether the SPARE-CVMs and related structural spatial patterns were CVM targeted. The other advice is that the author could use multiple linear regression with the five SPARE-CVMs as inputs and the CVM as outcomes, such analysis may assess the associations of the five SPARE-CVMs with CVM simultaneously.

To address the reviewer’s concern that SPARE-CVMs were CVM targeted, we performed multiple logistic regression analyses with binary CVM status (+/-) as the outcome variable and the 5 SPARE-CVMs as the predictors. As seen in the figure below, a unit increase in SPARE-

CVM was associated with a higher odds ratio of having a positive status in the target CVM, but not necessarily other co-occurring CVMs.

We have now included these results in the Appendix S-8, as Figure S-10C:

7. In Results-Evaluation of associations between SPARE-CVMs and cognitive performance: Similar concerns were on the site effects on the cognitive scores, which should be considered in the regression models.

We agree with the reviewer’s concerns regarding the widely different cognitive test batteries across studies and our regression models did indeed include confound correction for study differences. To have a meaningful investigation of cognition across the dataset, we chose only the cognitive tests that were implemented in multiple studies using the standardized protocols. We separated the cognitive investigations in UK Biobank from the rest of the dataset to minimize biased results due to the study’s large sample size and additionally because procedures in UK Biobank differed from the standardized cognitive protocols. For example, the digit symbol substitution test (DSST) was conducted over half the duration (60 seconds) in UKBioBank as opposed to the standard 120 seconds [27].

Our regression models incorporated study as a confounding covariate as described in Appendix 5.2:

“Multivariate linear regressions models predicting cognitive performance using SPARE-CVMs, while adjusting for the confounding covariates of study, age, sex, and number of years of education, were implemented on the rest of the dataset excluding UKBioBank cohort.”

8. In Results, the performance (accuracy, sensitivity and specificity in training and validation sets) of the SVM for each CVM should be provided. As showed in the validation set, the AUCs seemed around 0.7 for these CVMs, indicating a relatively low classification ability of the SVM models. The SPARE-CVMs derived from SVM classification with low performance may reduce the credibility of the SPARE-CVM and its clinical utility.

We updated our discussion to acknowledge that “our SPARE-CVMs show only moderate AUC levels around 0.7 for CVM classification. This study primarily aimed to develop quantitative markers for CVM-related sMRI patterns, rather than establish a diagnostic tool based on MRI.

The expression or manifestation of these CVM signatures at the time of MRI acquisition can be highly variable across participants due to factors like differing disease duration, severity management, and individual susceptibility to brain and cognitive changes. By quantifying the CVM-related neuroanatomical effects, SPARE-CVMs are intended to help identify CVM+ individuals with increased severity of brain alterations and not replace the far simpler and routine clinical diagnostic tools (e.g., blood pressure or weight measurements).”

Nevertheless, we now provide experimental validation of the improved performance with our model configuration compared to other ML methods (see R3Q3 above). This further suggests that ROI data may have a limited ability to distinguish CVMs compared to direct measurements of blood glucose or hemoglobin A1C.

We also include the following information in the updated Results section:

“Our ML configuration achieved better performance compared to other commonly employed ML models (**Appendix S4, Figure S-6C**) despite moderate receiver operating characteristic area under the curve (AUC) values for the training (and validation) datasets, which ranged between 0.64 (0.63) for SPARE-SM to 0.70 (0.72) for SPARE-OB.”

9. The treatment effect and follow up of the individuals on the different SPARE-CVM is warrant to be investigated in further studies.

We agree with the reviewer and acknowledge the need for future investigation regarding treatment effects in the Discussion section under limitations:

“The cross-sectional nature of our study and unavailability of information regarding disease or treatment duration across the sample pose a challenge in interpreting results related to medication status. ... Further longitudinal investigation is needed ...”

Minor concerns:

(1) The methods should be clearly described for a repeatability.

We have updated the methods. We also share the models on GitHub: https://github.com/stgovindarajan/spare_cvm_score and centiles of expected values from the harmonized dataset through the NiChart (<https://neuroimagingchart.com/>) platform. NiChart allows researchers to process their imaging data, derive the MUSE ROI segmentations used in the study, fit SPARE- models, and compare their results with the centile values from the reference set.

(2) A flow diagram regarding the data collection, processing and analysis etc. would make the study clear to readers.

We have now included a flow diagram.

Figure 1: Schematic overview of SPARE-CVM modeling.

(3) In Results, what's the exact meaning of $p < < 0.0001$? Corrected it as $p < 0.0001$?

Given our dataset's size, several statistical analyses yielded highly significant results. $p < < 0.0001$ is used to indicate tests where $-\log_{10}(p) > 50$, or $p < 1e^{-50}$.

(4) The inclusion of brain age gap seemed weird. Please provide the detailed reason. Please also provide the reason for SPARE-AD.

Brain age gap and SPARE-AD are well-established machine learning markers for identifying atrophy patterns associated with aging and AD. By incorporating these markers into our investigations, we aim to demonstrate their limited applicability for CVM-related atrophy patterns. We emphasize that SPARE-CVMs, as imaging-based phenotypic markers, provide a more nuanced understanding of how the brain is affected by cardiovascular changes compared to general measures of brain age.

(5) The contents in Table S-1 should be carefully modified and corrected.

We have updated Table S-1 and included additional citations.

(6) In Figure S-2, co-occurrence of CVM in the cross-validated training dataset should be also provided.

Figure S-2 has been updated to include the co-occurrence of CVM in the cross-validated training dataset and in the independent testing dataset (shown below).

Figure S-2: Co-occurrence of CVM in the cross-validated training dataset and external testing dataset.

References:

1. Shaw, L.M., et al., *Cerebrospinal fluid biomarker signature in Alzheimer's disease neuroimaging initiative subjects*. Ann Neurol, 2009. **65**(4): p. 403-13.
2. Soldan, A., et al., *Hypothetical Preclinical Alzheimer Disease Groups and Longitudinal Cognitive Change*. JAMA Neurol, 2016. **73**(6): p. 698-705.
3. Ewers, M., et al., *CSF biomarker and PIB-PET-derived beta-amyloid signature predicts metabolic, gray matter, and cognitive changes in nondemented subjects*. Cereb Cortex, 2012. **22**(9): p. 1993-2004.
4. Rowe, C.C., et al., *Amyloid imaging results from the Australian Imaging, Biomarkers and Lifestyle (AIBL) study of aging*. Neurobiol Aging, 2010. **31**(8): p. 1275-83.
5. Kamil, R.J., et al., *Vestibular Function and Beta-Amyloid Deposition in the Baltimore Longitudinal Study of Aging*. Front Aging Neurosci, 2018. **10**: p. 408.
6. Lopes Alves, I., et al., *Strategies to reduce sample sizes in Alzheimer's disease primary and secondary prevention trials using longitudinal amyloid PET imaging*. Alzheimers Res Ther, 2021. **13**(1): p. 82.
7. Joshi, A.D., et al., *Performance characteristics of amyloid PET with florbetapir F 18 in patients with alzheimer's disease and cognitively normal subjects*. J Nucl Med, 2012. **53**(3): p. 378-84.
8. Johnson, K.A., et al., *Florbetapir (F18-AV-45) PET to assess amyloid burden in Alzheimer's disease dementia, mild cognitive impairment, and normal aging*. Alzheimers Dement, 2013. **9**(5 Suppl): p. S72-83.
9. Villeneuve, S., et al., *Vascular risk and Abeta interact to reduce cortical thickness in AD vulnerable brain regions*. Neurology, 2014. **83**(1): p. 40-7.
10. Jang, H., et al., *Association of Glycemic Variability With Imaging Markers of Vascular Burden, beta-Amyloid, Brain Atrophy, and Cognitive Impairment*. Neurology, 2024. **102**(1): p. e207806.
11. Ye, B.S., et al., *Amyloid burden, cerebrovascular disease, brain atrophy, and cognition in cognitively impaired patients*. Alzheimers Dement, 2015. **11**(5): p. 494-503 e3.
12. Rabin, J.S., et al., *Association of beta-Amyloid and Vascular Risk on Longitudinal Patterns of Brain Atrophy*. Neurology, 2022. **99**(3): p. e270-e280.
13. Lo, R.Y., et al., *Vascular burden and Alzheimer disease pathologic progression*. Neurology, 2012. **79**(13): p. 1349-1355.
14. Hohman, T.J., et al., *Stroke risk interacts with Alzheimer's disease biomarkers on brain aging outcomes*. Neurobiology of aging, 2015. **36**(9): p. 2501-2508.
15. Newman, A.B., et al., *Dementia and Alzheimer's disease incidence in relationship to cardiovascular disease in the Cardiovascular Health Study cohort*. J Am Geriatr Soc, 2005. **53**(7): p. 1101-7.
16. Noale, M., F. Limongi, and S. Maggi, *Epidemiology of Cardiovascular Diseases in the Elderly*. Adv Exp Med Biol, 2020. **1216**: p. 29-38.
17. Muqtadar, H., F.D. Testai, and P.B. Gorelick, *The dementia of cardiac disease*. Curr Cardiol Rep, 2012. **14**(6): p. 732-40.
18. Pomponio, R., et al., *Harmonization of large MRI datasets for the analysis of brain imaging patterns throughout the lifespan*. Neuroimage, 2020. **208**: p. 116450.
19. Jack Jr, C.R., et al., *The Alzheimer's disease neuroimaging initiative (ADNI): MRI methods*. Journal of Magnetic Resonance Imaging: An Official Journal of the International Society for Magnetic Resonance in Medicine, 2008. **27**(4): p. 685-691.

20. *CARDIA BRAIN MRI SUBSTUDY*. Available from:
<https://www.cardia.dopm.uab.edu/images/more/pdf/mooy25/chapter12.pdf>.
21. Kraut, M.A., et al., *The impact of magnetic resonance imaging-detected white matter hyperintensities on longitudinal changes in regional cerebral blood flow*. *J Cereb Blood Flow Metab*, 2008. **28**(1): p. 190-7.
22. Resnick, S.M., et al., *One-year age changes in MRI brain volumes in older adults*. *Cereb Cortex*, 2000. **10**(5): p. 464-72.
23. Moghekar, A., et al., *Cerebral white matter disease is associated with Alzheimer pathology in a prospective cohort*. *Alzheimers Dement*, 2012. **8**(5 Suppl): p. S71-7.
24. Srinivasan, D., et al., *A comparison of Freesurfer and multi-atlas MUSE for brain anatomy segmentation: Findings about size and age bias, and inter-scanner stability in multi-site aging studies*. *Neuroimage*, 2020. **223**: p. 117248.
25. Erus, G. *MRISnapshot: Fast Visual QC of MRI Datasets*. Available from:
<https://cbica.github.io/MRISnapshot/>.
26. Feurer, M., et al., *Efficient and robust automated machine learning*. *Advances in neural information processing systems*, 2015. **28**.
27. Cockcroft, K., et al., *A cross-cultural comparison between South African and British students on the Wechsler Adult Intelligence Scales Third Edition (WAIS-III)*. *Front Psychol*, 2015. **6**: p. 297.

We thank the reviewers for their time and feedback. We have addressed Reviewer #3's questions (in blue) below and modified the manuscript as appropriate with tracked changes.

Reviewer #3 (Remarks to the Author):

The authors have addressed most of my concerns, however, some details should still be clarified as follows.

Regarding comment 1:

“(2) As shown in Figure S-1, it seemed that the author only removed the site effects, however, Table S1-1 listed MR scanners with heterogeneous field strength (including both 1.5T and 3T), vendors (GE, Siemens and Philips) and protocols (MPRAGE and non-MPRAGE sequences). I wonder whether the author include all these site-related confounding factors into consideration in the ComBat-GAM, as the Figure S-1 just provided the harmonization results across sites.”

The authors have provided examples that how they coded the “site” in the response. However, the absence of detailed information in the manuscript, supplementary materials,

We have included the following in Appendix Section 2.1 to clarify how site differences are coded for harmonization in the iSTAGING dataset.

“2.1 Harmonization of MUSE volumes

MUSE volumes in the iSTAGING dataset were harmonized using CombatGAM [1], a retrospective harmonization method used to mitigate scanner biases in the features (e.g., ROI volumes) derived from MRI. Scanner-level differences in the manufacturer, magnetic field strength, head coil, gradient and receiving coils, sequence specification, voxel size, etc. are modeled as site-specific differences in mean and variance across the pooled ROI volumes. Corrective site-level shift and scale parameters are then applied to the data to maintain relationships between data points within and across sites/batches.

The term ‘site’ or ‘batch’ here refers to the grouping of MRI data acquired using consistent scanner site and parameter settings during study data collection. Studies like ADNI were grouped into ‘batch’ or ‘site’ based on their study phase (i.e., ADNI-1, ADNI-2/ADNI-GO, ADNI-3) since the scans were prospectively standardized – i.e., the scanner field strength and sequences were adapted in a data-driven manner to ensure uniformity across the multicenter study [2]. In the case of studies like CARDIA, MRI data collection was performed across three different locations [3], and each is treated as a separate ‘batch’ or ‘site’ in ComBat-GAM. Similarly, BLSA-1.5T [4] and BLSA-3T [5] are treated as separate sites in the harmonization model, as the scanning parameters within site/phase were maintained for longitudinal consistency [6]. “

and the GitHub repository diminishes the credibility of the reported findings.

ComBat-GAM is a widely used method for harmonizing features from several imaging modalities [7, 8] and was a preprocessing tool in iSTAGING data consolidation. We did not

develop novel code or modify ComBat implementation for this paper, and hence, we did not include additional details regarding harmonization procedures in our GitHub repository. For more information on its implementation and use, we kindly refer the reviewer to other sources for ComBat family of harmonization methods [9, 10]. Harmonization is also available as a plugin on NiChart: [niCHART/niCHART/plugins/harmonization at main · CBICA/niCHART \(github.com\)](https://github.com/CBICA/niCHART) which users can upload their data to and compare SPARE- indices with the normative distribution.

“Additionally, some errors were observed in the Table S-1 regarding the protocols details, so I think the author may only take the site into consideration as the protocols they described were not correct (such as BIOCARD, if MPRAGE was applied, the TI was missing; ADNI, if MPRAGE was used for Philips, the TR and TI should not be same as Siemens, as these vendors had different TR definitions for MPRAGE sequences; some protocols provided FOV, some did not; some provided slice number, some did not).”

While the author has cited references that outline the protocols used, concerns remain unaddressed. Specifically, the OASIS study employs both 1.5T and 3T scanners, yet the protocols are identical, which seems implausible. Furthermore, the inversion time (TI) is set at a 20ms, and the definition of TD is unclear. Additionally, for the UKBB dataset, the flip angle (FA) and echo time (TE) parameters are absent.

We have updated Table S-1, including the scanner parameters listed for OASIS and UK Biobank, with additional citations from the source studies.

TD stands for delay time (or recovery) when the magnetization returns to equilibrium before the next inversion pulse, and no MRI signal is being generated [11]. We updated the legend for Table S-1 to include a list of abbreviations used in the table and their meanings:

“A subset of studies in the iSTAGING dataset are listed here to showcase the variations in sMRI acquisition protocols. The parameters listed here are representative, and additional information on how scanner variations are encoded in harmonization procedures can be found in Appendix S2.1. For more Abbreviations: FLAIR: Fluid-attenuated inversion recovery; MPRAGE: Magnetization-Prepared Rapid Gradient-Echo; PD: Proton density; SPGR: spoiled gradient echo; TSE: Turbo spin echo; ; TD: Delay time; TE: Echo time; TI: Inversion time; TR: Repetition time”

It is crucial to ensure consistency in these protocols across different sites. A thorough check and clarification of these details are necessary

We want to emphasize that large datasets using in “big data analytics” of neuroimaging, such as iSTAGING and ENIGMA [7], are assembled by combining data from various studies to overcome the challenges of small sample sizes and limited generalizability of study-specific inferences. Retrospective harmonization techniques, like ComBat, enable researchers to analytically and retroactively adjust for disparities in acquisition protocols when merging pre-existing diverse datasets. We expect that the original studies maintained internal consistency of

their acquisition protocols within sites. While we do not have control over acquisition parameters, we address the data quality concerns by using the MUSE segmentation tool, which is robust across multi-site datasets [12], and rigorous QC of derived MRI features.

Regarding comment 8:

“8. In Results, the performance (accuracy, sensitivity and specificity in training and validation sets) of the SVM for each CVM should be provided. As showed in the validation set, the AUCs seemed around 0.7 for these CVMs, indicating a relatively low classification ability of the SVM models. The SPARE-CVMs derived from SVM classification with low performance may reduce the credibility of the SPARE-CVM and its clinical utility.”

To facilitate a comprehensive evaluation of the models, it is essential to include the accuracy, sensitivity, and specificity metrics (preferably in the Supplementary materials). Although this study primarily aims to develop quantitative markers for CVM-related sMRI patterns, the performance of these models is paramount. This is because the reliability of the estimated quantitative markers is directly tied to the confidence in their predictive power.

We now include the following table in the appendix to address the reviewer's concern.

“Table S-4 Additional classification (CVM- and CVM+) performance metrics in the external validation dataset

A) Across the entire dataset, without excluding co-occurring CVMs					
SPARE-models	ROC-AUC	Balanced Accuracy	Accuracy	Sensitivity	Specificity
Hypertension	0.709	0.652	0.651	0.644	0.661
Hyperlipidemia	0.709	0.650	0.634	0.707	0.593
Smoking	0.633	0.595	0.597	0.591	0.599
Obesity	0.716	0.655	0.693	0.560	0.749
Diabetes	0.706	0.657	0.674	0.638	0.676
B) In a subset of participants with no comorbidities (i.e., excluding participants with CVM+ in more than one CVM)					
SPARE-models	ROC-AUC	Balanced Accuracy	Accuracy	Sensitivity	Specificity
Hypertension	0.698	0.637	0.643	0.547	0.728
Hyperlipidemia	0.708	0.639	0.665	0.592	0.686
Smoking	0.613	0.580	0.650	0.463	0.697
Obesity	0.698	0.625	0.724	0.478	0.772
Diabetes	0.692	0.673	0.759	0.585	0.761

“

References:

1. Pomponio, R., et al., *Harmonization of large MRI datasets for the analysis of brain imaging patterns throughout the lifespan*. Neuroimage, 2020. **208**: p. 116450.
2. Jack Jr, C.R., et al., *The Alzheimer's disease neuroimaging initiative (ADNI): MRI methods*. Journal of Magnetic Resonance Imaging: An Official Journal of the International Society for Magnetic Resonance in Medicine, 2008. **27**(4): p. 685-691.
3. *CARDIA BRAIN MRI SUBSTUDY*. Available from: <https://www.cardia.dopm.uab.edu/images/more/pdf/mooy25/chapter12.pdf>.
4. Kraut, M.A., et al., *The impact of magnetic resonance imaging-detected white matter hyperintensities on longitudinal changes in regional cerebral blood flow*. J Cereb Blood Flow Metab, 2008. **28**(1): p. 190-7.
5. Resnick, S.M., et al., *One-year age changes in MRI brain volumes in older adults*. Cereb Cortex, 2000. **10**(5): p. 464-72.
6. Moghekar, A., et al., *Cerebral white matter disease is associated with Alzheimer pathology in a prospective cohort*. Alzheimers Dement, 2012. **8**(5 Suppl): p. S71-7.
7. Radua, J., et al., *Increased power by harmonizing structural MRI site differences with the ComBat batch adjustment method in ENIGMA*. Neuroimage, 2020. **218**: p. 116956.
8. Horng, H., et al., *Improved generalized ComBat methods for harmonization of radiomic features*. Sci Rep, 2022. **12**(1): p. 19009.
9. Chen, A.A., *ComBatFamily GitHub Repository*. 2024.
10. Radua, J. *ComBat functions for ENIGMA sMRI in R 2020* [cited 2024 Oct]; Available from: http://enigma.ini.usc.edu/wp-content/uploads/combat_for_ENIGMA_sMRI/combat_for_ENIGMA_sMRI.R.
11. Mugler, J.P., 3rd and J.R. Brookeman, *Rapid three-dimensional T1-weighted MR imaging with the MP-RAGE sequence*. J Magn Reson Imaging, 1991. **1**(5): p. 561-7.
12. Doshi, J., et al., *MUSE: MUlti-atlas region Segmentation utilizing Ensembles of registration algorithms and parameters, and locally optimal atlas selection*. Neuroimage, 2016. **127**: p. 186-195.